# Calorie restriction increases insulin sensitivity to promote beta cell homeostasis and longevity in mice

Cristiane dos Santos [1,5], Amanda Cambraia [1,5], Shristi Shrestha[1], Melanie Cutler[1], Matthew Cottam [1], Guy Perkins [2], Varda Lev-Ram [2], Birbickram Roy [3], Christopher Acree [1], Keun-Young Kim[2], Thomas Deerinck[2], Danielle Dean[1], Jean Philippe Cartailler [1], Patrick E. MacDonald[3], Martin Hetzer [4], Mark Ellisman [2] & Rafael Arrojo e Drigo [1] ✉

Caloric restriction (CR) can extend the organism life- and health-span by improving glucose homeostasis. How CR affects the structure-function of pancreatic beta cells remains unknown. We used single nucleus transcriptomics to show that CR increases the expression of genes for beta cell identity, protein processing, and organelle homeostasis. Gene regulatory network analysis reveal that CR activates transcription factors important for beta cell identity and homeostasis, while imaging metabolomics demonstrates that beta cells upon CR are more energetically competent. In fact, high-resolution microscopy show that CR reduces beta cell mitophagy to increase mitochondria mass and the potential for ATP generation. However, CR beta cells have impaired adaptive proliferation in response to high fat diet feeding. Finally, we show that long-term CR delays the onset of beta cell aging hallmarks and promotes cell longevity by reducing beta cell turnover. Therefore, CR could be a feasible approach to preserve compromised beta cell structure-function during aging and diabetes.

Aging is associated with loss of normal cell and tissue function, which are linked to degenerative and metabolic diseases such as Alzheimer's and diabetes during old age[1]. Tissues composed of largely post-mitotic cells are expected to be significantly impacted by ageing because most cells in these tissues can be remarkably long-lived and as old as the organism itself[2–4]. Because of their longevity, long-lived cells (LLCs) are under immense pressure to maintain their normal cell function for long periods of time to sustain organ function. However, how LLCs achieve and maintain structure and function during adulthood, and whether aging-associated deficits in LLC function in old organisms can be reversed to restore normal tissue function remains largely unknown.

Using a combination of in vivo stable isotope labeling of mammals (SILAM), correlated and high-resolution electron and multi-isotope mass spectrometry (called MIMS-EM), and protein mass spectrometry pipelines, we have identified the tissue distribution of LLCs and of long-lived protein (LLP) complexes in post-mitotic cells[4–8]. These studies identified that different endocrine cell types in the mouse pancreas are LLCs, including up to 60% of all insulin-secreting beta cells[4,9,10]. Beta cells secrete insulin in response to increases in blood glucose levels to sustain normal glucose homeostasis for an entire lifetime[11]. Several studies have investigated the impact of aging on beta cells and established that aging beta cells have compromised expression of transcription factors (TFs) and re-organization of gene

[1]Vanderbilt University, Department of Molecular Physiology and Biophysics, Nashville, La Jolla, TN, USA. [2]National Center for Imaging and Microscopy Research, University of California San Diego, La Jolla, CA, USA. [3]Department of Pharmacology and Alberta Diabetes Institute, University of Alberta, Edmonton, AB, Canada. [4]Institute of Science and Technology Austria (ISTA), Vienna, Austria. [5]These authors contributed equally: Cristiane dos Santos, Amanda Cambraia. ✉e-mail: r.drigo@vanderbilt.edu

regulatory networks (GRNs) that maintain beta cell identity[12–14], accumulation of islet fibrosis, inflammation, and ER stress[12,15], reduced KATP channel conductance[16], loss of coordinated beta cell calcium dynamics[17], impaired beta cell autophagy and accumulation of beta cell DNA damage, and/or beta cell senescence[12,14,18,19]. When combined with an unhealthy lifestyle during old age, these aging signatures could pre-dispose beta cells to failure and lead to type 2 diabetes (T2D) onset[20].

Caloric restriction (CR) and CR-mimicking approaches (e.g., time-restricted feeding (TRF)) can prolong organismal longevity and delay aging from yeast to non-human primates[21–26]. These beneficial effects are associated with improved glucose homeostasis due to prolonged fasting and enhanced peripheral insulin sensitivity, enhanced insulin signaling, lower adiposity, enhanced mitochondrial homeostasis and lower ATP generation, increased autophagy, enhanced protein homeostasis and reduced ER stress and inflammation[21–29]. In the pancreas, 20–40% CR or CR achieved via TRF are linked to lower islet cell mass in lean mice[21], whereas in pre-diabetic mice, CR restores normal beta cell secretory function, identity, and preserves beta cell mass[30–33] in a process dependent on activation of beta cell autophagy via *Beclin-2*[34]. In patients with a recent T2D diagnosis, the efficacy of extreme CR (average of 835 kcal/day or >50% CR based on a 2000 kcal/day diet) to reverse T2D depends on the capacity of beta cells to recover from previous exposure to a T2D metabolic state[35]. However, how beta cells adapt during CR, and whether CR can delay the hallmarks of beta cell aging remains largely unknown.

We investigated these questions by exposing adult male mice to mild CR (i.e., 20% restriction) for up to 12 months and applied comprehensive in vivo and in vitro metabolic phenotyping of beta cell function followed by single-cell multiomics and multi-modal high-resolution microscopy pipelines. Our data reveals that CR reduces the demand for beta cell insulin release necessary to sustain euglycemia by increasing peripheral insulin sensitivity. Ad-libitum (AL) consumption of a diet with reduced caloric intake failed to trigger a similar phenotype, thus indicating that CR and CR-associated fasting periods are required for this adaptive metabolic response. During CR, the transcriptional architecture of beta cells is re-organized to promote a largely post-mitotic and long-lived phenotype with enhanced cell homeostasis and mitochondrial structure-function. This is associated with reduced onset and/or expression of beta cell aging and senescence signatures. When exposed to a high-fat diet (HFD), CR beta cells upregulate insulin release, however they have a compromised adaptive response due to limited cell proliferation resulting in reduced beta cell mass.

Therefore, our results provide a molecular footprint of how CR modulates adult beta cell function and insulin sensitivity to promote beta cell longevity and delay aging in mice.

## Results
### Calorie restriction (CR) modulates beta cell function in vivo
We exposed 8-week-old FVB or C57/BL6 mice to 20%, CR for 2 months, and quantified in vivo glucose homeostasis mechanisms (Fig. 1A). As expected, CR mice consumed 80% of the total daily kilocalories (kcal) consumed by control mice maintained in an ad libitum (AL) diet (Fig. S1A and S2A, B). After 2 months, CR mice did not experience significant body weight gain due to a reduction in fat mass and adiposity, and maintained lean body mass (Fig. 1B and S1B, S2A–D). In contrast, mice exposed to HFD for 2 months consumed more calories and became obese (Fig. 1B, C and S1A).

To quantify the impact of CR on beta cell insulin release and glucose homeostasis in vivo, we performed a series of oral mixed-meal tolerance tests (MTT) using a high-carbohydrate nutrient shake (Ensure, ~2% of the daily kcal consumption, see methods for details), or an oral glucose tolerance test (2 mg/kg). This revealed that CR mice have improved glucose tolerance compared to AL mice, whereas HFD

mice were glucose intolerant—as expected (Fig. 1E, Figs. S1C and S2E). However, fasting glucose levels were not different between AL and CR mice (CR ($n = 26$) $105.4 \pm 14.81$ mg/dL versus AL ($n = 24$) $114.0 \pm 19.34$ mg/dL, $p = 0.0834$). Serum insulin measurements before and during the MTT revealed that CR beta cells secrete ~50% less insulin than AL beta cells, despite having a similar stimulated insulin secretory capacity (Fig. 1F, G, Figs. S1E and S2G). Additional experiments where AL or CR mice were pre-treated with the GLP1 receptor agonist Liraglutide (400 µg/kg body weight) revealed improved glucose tolerance and insulin release in both groups, with AL mice sustaining higher serum insulin levels after 30 min (Fig. S1K–M). Together, these results suggest that CR does not impair beta cell-stimulated insulin secretion mechanisms in vivo. Instead, CR beta cells enjoy a lower requirement for insulin release to maintain euglycemia due to enhanced peripheral insulin sensitivity (Fig. 1H–J). Of note, no changes in hepatic insulin degradation enzyme (IDE) activity, alpha or beta cell mass, or circulating glucagon levels (AL ($n = 11$): $2.473 \pm 1.473$ versus CR ($n = 12$) $2.896 \pm 1.240$ pM) were observed after 2 months on diet ((Fig. S1G–F). In contrast, CR failed to enhance glucose homeostasis or modify beta cell function in female mice despite significant changes to body weight and lower body fat composition (Fig. S3A–L).

To investigate beta cell function and insulin release mechanisms in vitro, we performed dynamic glucose-stimulated insulin secretion assays (GSIS) using isolated islets from male mice exposed to AL, CR, and HFD for 2 months. Surprisingly, no significant differences in basal and/or glucose-stimulated insulin release or islet insulin content were observed between diet groups (Fig. S1H, I). However, CR beta cells had reduced insulin release when challenged with high glucose in combination with the phosphodiesterase inhibitor 3-isobutyl-1-methylxanthine (IBMX). While this observation may indicate that CR beta cells have impaired glucose-stimulated cAMP generation, these results are likely due to in vitro aspects that fail to recapitulate the in vivo beta cell environment and GLP1R signaling (Fig. S1K–M). Together, our experiments show that CR improves peripheral insulin sensitivity and glucose tolerance in male mice, which decreases the demand for beta cell insulin secretion required to maintain normoglycemia without compromising insulin release mechanisms. In contrast, female mice appear to be resistant to most of the effects of CR, likely due to sex-specific differences in glucose metabolism[21].

### Calorie dilution is insufficient to modulate beta cell function in vivo
During CR, mice experience periodical events of feeding intercalated by long fasting periods that are necessary for the beneficial metabolic effects of CR[29]. First, to determine if an acute exposure to CR (i.e., 15 days) would be sufficient to affect beta cell insulin release, we accessed glucose homeostasis and beta cell insulin secretion after 2-, 4-, or 6-weeks of AL or CR feeding. This revealed that at least 6 weeks of CR feeding is required to significantly improve glucose tolerance and lower beta cell insulin release (Fig. S2I–K). Next, to test if reducing calorie intake alone (without prolonged fasting periods) can modulate beta cell function in vivo, we exposed mice to AL, CR, or food pellets with 20% cellulose. This approach dilutes the amount of bioavailable energy in the diet while maintaining AL access to food and minimize fasting periods[29]. As expected, CR mice were leaner with improved glucose homeostasis and lower circulating insulin levels (Fig. 1K–N). In contrast, mice fed a 20% diluted chow (DL) had similar glucose homeostasis and circulating insulin levels as AL mice, despite being relatively leaner (Fig. 1K–N). Notably, DL mice consumed ~18% more food over time than AL mice to maintain normal levels of digested food energy (Fig. S3M–Q), thus showing that DL-fed mice adjust their feeding behavior to maintain normal energy levels. Together, this data indicates that sustained CR is required to trigger a functional beta cell adaptation, and that glucose homeostasis mechanisms are not affected by calorie dilution alone.

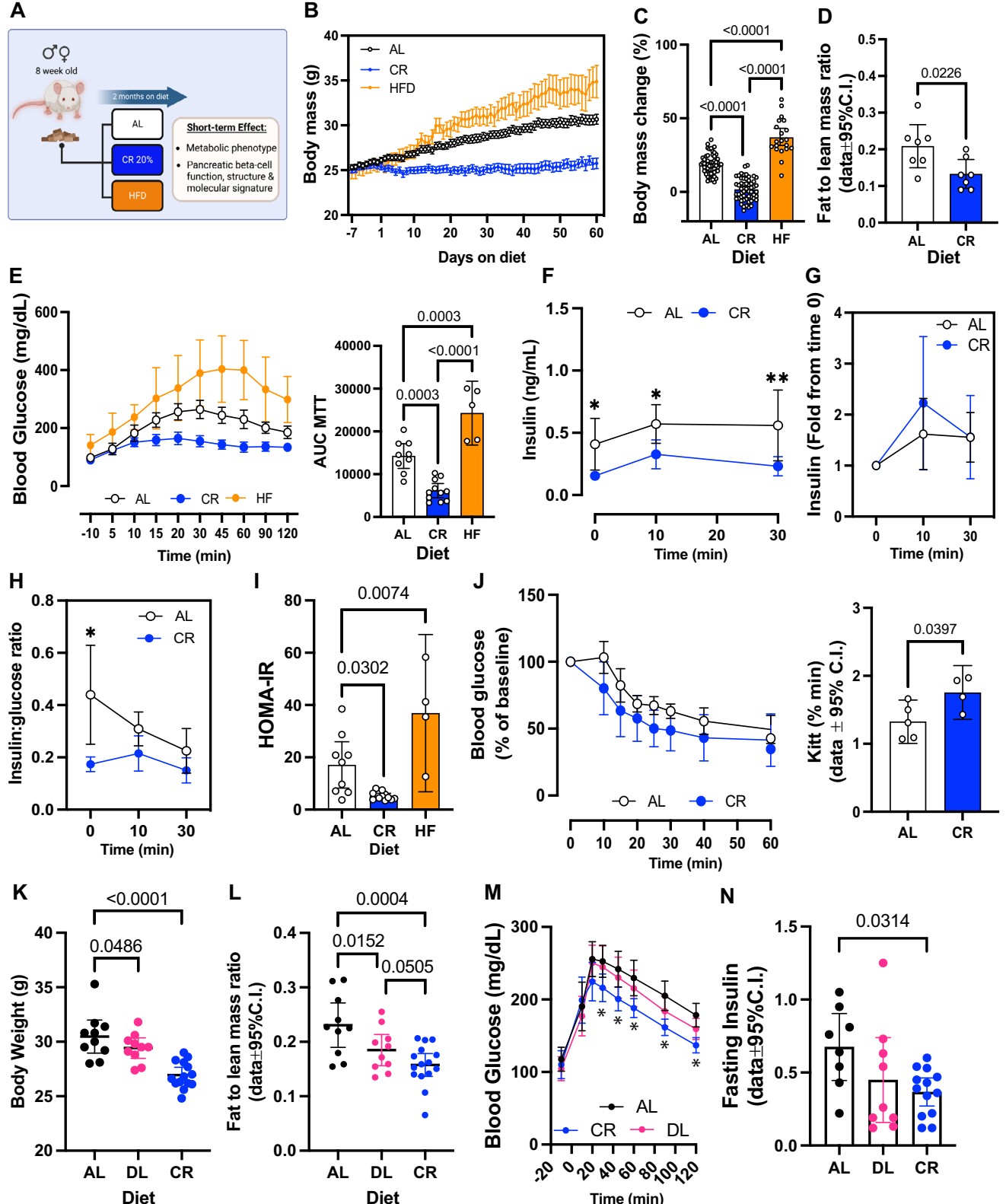

## CR beta cells have impaired adaptive replication response to HFD feeding

TRF, CR, or dieting can recover compromised beta cell function during diabetes in mice and humans[32,33,35]. However, successive oscillations in body weight led to compromised glucose homeostasis due to impairment of beta cell function[36]. To determine whether CR exposure protects the glucose homeostasis mechanisms during times of

metabolic stress, we fed AL or CR mice with HFD for 2 months and monitored glucose clearance and beta cell insulin secretion in vivo. Of note, re-introduction of CR mice to AL access to chow diet (CR-AL group) leads to significant body weight gain and restores mouse body fat mass content, glucose tolerance, and insulin release, thus indicating that the impact of CR on beta cell function is reversible (Fig. 2A–F). Strikingly, CR mice exposed to HFD (CR-HF group) become obese,

**Fig. 1 | Short-term caloric restriction (CR) improves glucose homeostasis in male mice. A** Schematic diagram of mice subjected to ad libitum (AL), 20% CR, or HFD for 2 months starting at 8 weeks of age. **B** Mouse body mass over 2 months on AL, HFD, or CR diet. **C** Body mass change after 2 months on diet. **D** Ratio between fat mass and lean mass after 2 months on diet AL, HFD, or CR diet. **E** Blood glucose levels during the meal tolerance test (MTT) after 2 months on diet and respective area under-curve (AUC) measurements. **F** Insulin levels during the MTT and **G** the respective fold change from baseline values. **H** Ratio between the insulin and glucose values obtained during the MTT. **I** HOMA-IR calculated from fasting glucose and insulin values after 2 months on diet. **J** Blood glucose values obtained during an intraperitoneal insulin tolerance test (ipITT) and the respective decay of the glucose rate per minute (kITT). **K** Mouse body mass after 2 months on AL, CR, or 20% diluted diet (DL). **L** Ratio between fat mass and lean mass of AL, CR, or DL mice. **M, N** Blood glucose levels and circulating insulin levels of AL, CR, and DL mice

during a meal tolerance test (MTT) after 2 months on diet. Each dot represents the mean measurement for individual mice. **B, C** $n = 60$ AL, $n = 75$ CR, and $n = 20$ HF mice per group; **D** $n = 7$ mice per diet group; **E–I** $n = 9$ AL, $n = 11$ CR, and $n = 5$ HF mice per group; **J** $n = 5$ mice per diet group; **K–M** $n = 10$ AL, $n = 10$ DL, and $n = 15$ CR mice per group; **N** $n = 8$ AL, $n = 9$ DL, and $n = 13$ CR mice per group. $P$ values for all significantly different comparison are shown. **F, H** the asterisks indicate $*p < 0.05$, $**p < 0.01$. Statistical analysis was conducted using one-way ANOVA with Tukey's post-hoc test **C, E, I**, or followed by Dunnett's post-hoc test (**K** and **N**) or Benjamini, Krieger, and Yekutieli's post-hoc test (**L**). Data on (**F–H, M**) were analyzed with two-way ANOVA with Sidak's post-hoc test or Tukey's post-hoc test. For two groups comparison, unpaired two-tailed Student's $t$ test was performed (**D, J**). All data presented as mean ± 95% confidence intervals (C.I.). **A** was created with BioRender.com. and released under a Creative Commons Attribution-Non-Commercial NoDerivs 4.0 International license.

hyperglycemic, and as glucose intolerant as control HFD-fed mice (AL-HF group, Fig. 2A–D). Moreover, CR-HF mice have impaired meal-stimulated beta cell insulin release despite similar insulin sensitivity as AL-HF mice (Fig. 2E, F).

To investigate the underlying molecular signature of CR-HF beta cells, we performed bulk RNA-seq of isolated islets from AL-HF and CR-HF mice. We identified a total of $n = 555$ genes differentially regulated (Supplementary Data 1), with $n = 495$ genes down-regulated in CR-HF mice. Pathway enrichment analysis revealed that these genes were linked to chromatin remodeling and cell cycle progression (*Atrx, Brd4/9*), tissue morphogenesis (*Mef2c, Foxp1*), and pancreas cell development (*Nkx6-1, Sox9*) and function (*Cela2a, Cpa1/2*) (Fig. 2G). This data hinted that CR beta cells could have a compromised adaptive replication response to HFD feeding; to address this possibility, we quantified the beta cell mass in CR-HF and AL-HF mice and found a significantly lower beta cell mass in CR-HF mice (Fig. 2H). Our experiments demonstrate that CR beta cells are neither "protected" nor do they have enhanced glucose homeostasis in response to HFD feeding. In contrast, CR beta cells have suppressed activation of transcriptional programs and proliferation necessary for successful expansion of the beta cell mass in response to HFD.

## CR alters beta cell epigenetic and transcriptional heterogeneity

Given that CR triggered significant changes to glucose homeostasis and beta cell insulin secretion in vivo, while modulating the transcriptional landscape of CR islets after HFD feeding (Figs. 1 and 2), we applied single nucleus (sn) transposase-accessible chromatin with sequencing (ATAC)- and mRNA sequencing (called snMultiome-seq, 10x Genomics) to investigate the transcriptional architecture underlying the adaptation of beta cells to CR or HFD after 2 months on diet.

Single nuclei were isolated from frozen islet pellets from mice exposed to AL, CR, or HFD diet for 2 months (Fig. 3A). We analyzed a total of 16,555 islet nuclei to identify major islet cell types using well-established islet cell gene expression and chromatin accessibility patterns (Fig. 3B–F and S4A–F)). Here, pseudo-bulk differential gene expression analysis of AL, CR, and HFD beta cell transcriptomes revealed that CR beta cells have upregulation of several beta cell identity genes, including both insulin genes (*Ins1, Ins2*), amylin (*Iapp*), the insulin processing enzyme *Pcsk1n*, the glucose-6 phosphatase enzyme *G6pc2*, the beta cell TF *Nkx6-1*, and downregulation of the incretin receptor *Gipr* (*Glp1r* was not detected) and of the carbohydrate-responsive transcriptional regulator *Mlxipl* (also known as *ChREBP*) (Fig. 3G, Supplementary Data 2). In contrast, CR had a limited effect on beta cell chromatin accessibility ($n = 143$ differentially modulated chromatin with $p$ val < 0.05), where most changes were associated with genes containing TF motifs mapped to circadian rhythm- (*Arntl/Bmal, Dbp, Tef,* and *Hlf*) and nutrient-dependent regulators of autophagy (*Tfeb* and *Tfe3*) (Fig. 2H, Supplementary Data 3).

Beta cells with higher levels of *Arntl/Bmal* have been associated with a more functional phenotype with increased expression of beta cell identity markers such as *Pdx1*[37,38]. Immunohistochemistry (IHC) and confocal microscopy of islets from CR mice confirmed that ~80% of all CR beta cells have higher levels of *Arntl/Bmal* and Pdx1 in situ after 2 months on diet (Fig. S4G). Moreover, pathway enrichment analysis of differentially expressed genes and enriched chromatin regions in CR beta cells revealed enrichment of pathways linked to beta cell identity and function, oxidative phosphorylation (OxPhos), mitophagy, protein processing, degradation, and transport through the endoplasmic reticulum (ER) and Golgi apparatus (Fig. 3I). In contrast, HFD beta cells showed reduced expression of genes associated with beta cell function (i.e., *Fos, Fosb, Jun,* and the incretin receptor *Gipr*), and increased expression of stress- and diabetes-associated genes (*Fkbp5* and *Glis3*, respectively) (Fig. 3G–I).

Next, we identified three different beta cell transcriptional states (beta cell states 0, 1, and 2) and their marker genes and chromatin accessibility profiles (Fig. 3J, K, Fig. S4H–J). This analysis shows that CR caused a shift in the transcriptional heterogeneity of beta cells, where CR mouse islets had ~2x more beta cells in state 2 (versus AL and HFD islets (Fig. 3J)). Beta cells in this transcriptional state have higher expression of beta cell identity and function (*Ins1, Ins2, Mafa*), oxidative and anaerobic carbohydrate metabolism (*Pkm, Idh2, Acly*) and mitochondrial structure and function (Cytochrome C genes (*Cox4-to-8*), *Vdac1, Tomm20,* and *Tomm70a*), autophagy (*Lamp1, Cltc, p62/Sqstm1, Ulk1, Ctsb*), lipid metabolism (*Hadh*), amino-acid transport and metabolism (*Slc7a2, Slc38a2*), protein folding and response to ER stress (*Hspa5, Atf6, Creb3l2*) and DNA damage (*Nme1*) genes (Fig. 3K, Figs. S5A–E and S8C–F, Supplementary Data 4). Moreover, state 2 beta cells had lower expression of the rate-limiting enzyme glucokinase (*Gck*) and increased expression of glucose-6-phosphatase catalytic subunit 2 (*G6pc2*), which could suggest that these cells have lower glycolytic flux and increased glucose excursion[39]. However, this does not appear to be the case since CR beta cells have appropriate glucose-stimulated insulin release dynamics (Fig. 1 and Fig. S1).

Together, these results indicate that CR triggers significant transcriptional and chromatin accessibility changes in most beta cells that enhance cell identity, nutrient metabolism, circadian rhythm regulation, and ER, mitochondria, and protein homeostasis mechanisms.

## CR reprograms beta cell GRNs

Beta cell development, maturation, and ageing processes are characterized by modulation of gene regulatory networks (GRNs) that control beta cell gene expression patterns and heterogeneity[12,40,41]. Given that CR has limited effects on beta cell chromatin accessibility (Fig. 3, Figs. S4 and 5), we wanted to test if the transcriptional phenotype of CR beta cells was largely due to re-organization of beta cell GRNs. To test this hypothesis, we applied the TF inference tool SCENIC[42,43] to scan the transcriptional signatures of individual beta

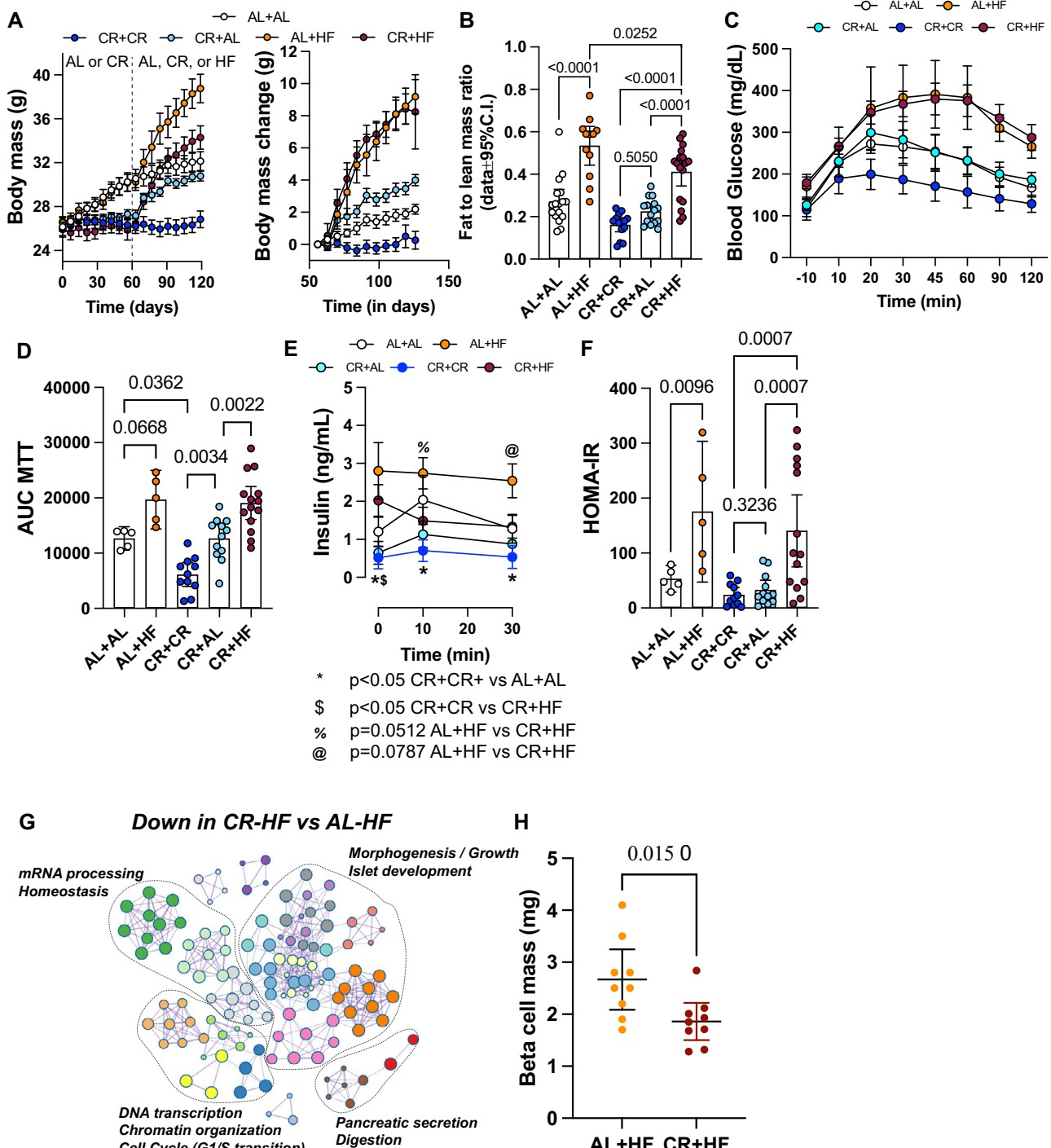

**Fig. 2 | CR disrupts beta cell adaptive response during HFD. A** Mouse body mass and body mass change over the course of 4 months, before and after switching diets. Mice were placed on AL or CR diets for 2 months and then shifted to AL (CR-AL) or HFD (AL-HF or CR-HF) for another 2 months. Two groups of AL (AL-AL) or CR (CR-CR) mice were maintained on their original diets as controls. **B** Ratio between fat mass and lean mass at the end of the diet switch experiment. **C, D** Blood glucose levels during the meal tolerance test (MTT) and area under-curve (AUC) measurements at the end of the diet switch experiment. **E** Insulin levels during the MTT shown in (**C**). All *p* values for select comparisons are listed at the bottom. **F** HOMA-IR was calculated from fasting glucose and insulin values at the end of the diet switch experiment. **G** Bulk-RNA-seq pathway enrichment analysis graph displaying interconnected nodes representing pathways downregulated in CR-HF mice versus

AL-HF. **H** Pancreas beta cell mass in AL-HF versus CR-HF mice at the end of the diet switch experiment. *P* values for all significantly different comparison are shown, and each dot represents the mean measurement for individual mice. **A** *n* = 29 AL-AL, *n* = 19 AL-HF, *n* = 24 CR-CR, *n* = 40 CR-AL, and *n* = 27 CR-HF mice per group; **B** *n* = 17 AL-AL, *n* = 12 AL-HF, *n* = 14 CR-CR, *n* = 15 CR-AL, and *n* = 17 CR + HF mice per group; **C–F** *n* = 5 AL-AL, *n* = 5 AL-HF, *n* = 11 CR-CR, *n* = 12 CR-AL, and *n* = 14 CR-HF mice per group; **H** *n* = 9 mice per group. Statistical analysis was conducted using one-way ANOVA with Tukey's post-hoc test (**B**, **D**), or followed by Benjamini, Krieger, and Yekutieli's post-hoc test (**F**). **E** two-way ANOVA with Benjamini, Krieger, and Yekutieli's post-hoc test was performed. For two groups comparison, unpaired two-tailed Student's *t* test was performed (**H**). All data presented as mean ± 95% confidence intervals (C.I.).

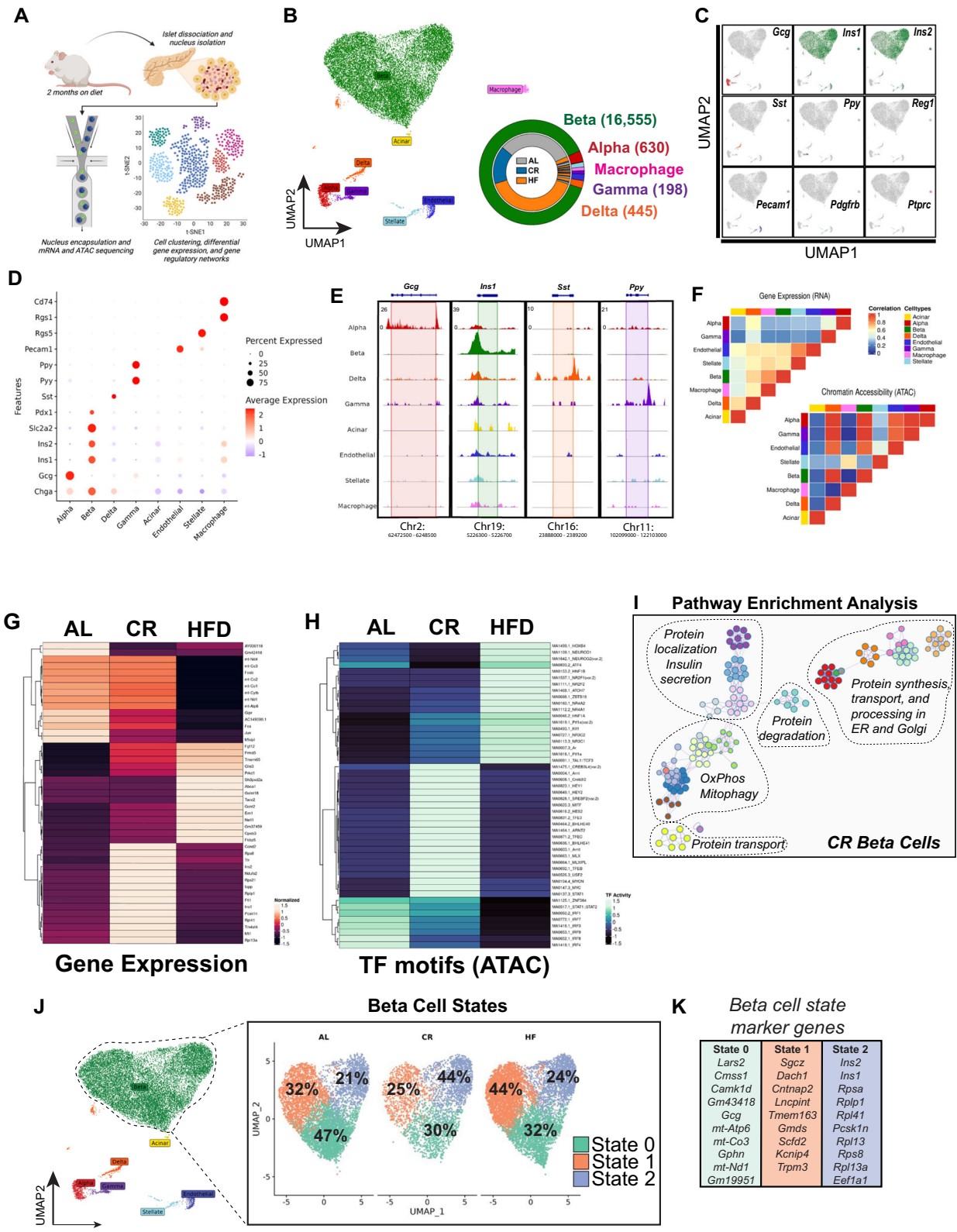

cells to infer and assign TF-gene linkages using a known TF motif database to create cell type-specific GRNs[12,42,43].

We applied SCENIC to our single beta cell snMultiome-seq dataset ($n = 16{,}555$ cells in total) from AL, CR, and HFD mice after 2 months on diet and identified a total of 242 TF enriched in all beta cells, including beta cell identity TFs *Pdx1*, *Nkx-6.1*, *Mafa*, and *Foxa2a* (Fig. 4A, Supplementary Data 5). Next, we determined the overall

degree of correlation in the inferred activity of each TF to identify modules of co-regulated TFs (as we have previously shown in ref. 12). This approach revealed the expected and positive correlation in the activation of beta cell TFs (*Pdx1*, *Mafa*, and *Foxa2*, (Fig. 4B)[44] and identified distinct sets of co-regulated beta cell TFs, including the insulin-sensitive *Foxo1*, the ER homeostasis and stress response *Atf6*, the beta cell identity *Nkx6.1* and the carbohydrate-responsive

**Fig. 3 | Single-cell multiome sequencing of islets reveals diet-specific changes to beta-cell heterogeneity. A** Schematic representation of the workflow used for single nuclei multiome sequencing (ATAC + RNA) of pancreatic islets isolated from AL, CR and HFD mice after 2 months on diet. **B** Uniform Manifold Approximation and Projection (UMAP) of the integrated transcriptome and chromatin dataset. Data from a total of $n = 18,741$ islet cells from $n = 2$ mice per diet group. Different cell types are indicated by various colors and labels. Inset, donut plot with the total number of cells in each cell cluster. **C** UMAPs show the expression of cell marker genes. **D** Dot plot with relative expression of marker genes in alpha, beta, delta, gamma, acinar, endothelial, stellate, and macrophage cell clusters. **E** Genome tracks showing ATAC peaks for hormone markers: Gcg (alpha cell), Ins1 (beta cell), Sst (delta cell), and Ppy (gamma cell). **F** Heatmap showing the correlation between gene expression (RNA-seq) and chromatin accessibility (ATAC-seq) across the identified cell types. **G** Heatmap with the top differentially expressed genes (RNA-seq) and **H** top ATAC-seq peaks in transcriptional factors (TF) motifs in beta cells from AL, CR, and HFD mice. **I** Annotated node map with pathway enrichment analysis of differentially expressed genes and TF motifs in beta cells from CR mice. Analysis was performed using Metascape with an FDR < 0.05. **J** UMAP projection of beta cells and identified subclusters from AL, CR, and HFD mice. Beta cell populations were defined according to distinct transcriptional states. **K** Top marker genes from each beta-cell state. **A** was created with BioRender.com. and released under a Creative Commons Attribution-Non-Commercial NoDerivs 4.0 International license.

---

*Mlxipl/ChREBP*, and two large groups of TFs associated with control of gene transcription mechanisms and cell development (*Sox9, Klf3/4, Runx1-3*) (Fig. 4B).

Next, we used SCENIC to reconstruct diet-specific beta cell GRNs and to identify modules of TFs with similar activity patterns (Fig. 4C, Fig. S6A–D) and found that CR and HFD are associated with a significant reprograming of beta cell GRNs and TF activities (Fig. 4C). Compared to AL beta cells, the CR beta cell GRN was characterized by increased activation and/or coordinated activity of TFs involved in beta cell identity and function (*Mafa, Nkx6-1, Foxo1*), protein folding and ER homeostasis (*Creb3l2, Atf6*), transcriptional regulation of mitochondrial genes (*Gabpa/Nrf2* and *Gabpb1*), NAD metabolism (*Sirt6*), and glucocorticoid signaling (*Nr3c1*) genes, which coincides with increased gene expression of some of these markers in CR beta cells (Figs. 3, 4, and S6D). In contrast, the GRN in HFD beta cells had increased activity of the glucose-responsive TF *Mlxipl/ChREBP*, which is required for glucose-stimulated beta cell proliferation and diabetes[45,46], and of the beta cell identity TF *Mafa* (Fig. 4C). Notably, the network of genes targeted by *Mafa* in CR and HFD beta cells is not similar: while in CR beta cells *Mafa* is linked to autophagy and ER homeostasis genes *Beclin1* and *Calmodulin1*, respectively, in HFD beta cells *Mafa* is linked to master regulators of beta cell proliferation (*Ccnd1, Cdc42*) and ER stress response (*Atf4*) (Fig. S6E, Supplementary Data 6, 7). Similar results were observed for the beta cell proliferation TF *Foxp1*, and the protein homeostasis TF *Crebl2* (Fig. S6E). Importantly, this beta cell TF target "reprogramming" was reflected in the gene expression pattern of genes associated with *Mafa* and *Atf6*. Here, these TFs in CR beta cells were linked to the increased expression of several beta cell identity and protein homeostasis genes (Figs. 3 and 4), including the beta cell-specific ER chaperone *Ero1lb*, the alpha and beta subunits of the ER translocation channel Sec61 (*Sec61a1, Sec61b*), the regulator of intra-Golgi transport *Arf1*, the autophagy marker *Lamp1*, and beta cell identity and function genes *Ucn3, Chga*, and *Nkx6-1* (Fig. 4D).

Therefore, CR promotes beta cell identity and homeostasis by upregulating the expression of beta cell maturation, homeostasis, and function genes. This transcriptional program is linked to the transcriptional activity of *Mafa/Nkx6.1* and of other TFs involved in the maintenance of beta cell protein homeostasis and survival mechanisms (e.g., *Atf6*[47]).

### Imaging mass spectrometry defines the spatial metabolomic landscape of CR islets

Our data indicates that CR changes beta cell function via modulation of beta cell identity, nutrient metabolism, and organelle homeostasis pathways (Figs. 1–4). Next, we decided to investigate the metabolic profile of in situ islets using imaging mass spectrometry (called MALDI-MS) to quantify the relative abundance and spatial profile of metabolites in islets from AL and CR mice after 2 months on diet. We chose this approach to leverage MALDI-MS's capabilities to quantify metabolic heterogeneity within tissue compartments while minimizing potential changes in cell metabolism that could occur during the islet isolation process. Here, fresh mouse pancreases were snap-frozen using a liquid nitrogen bath and placed in histology cassettes prior to MALDI-MS metabolite imaging and data analysis (see methods for details) (Fig. 5A, B and S7A). Consecutive sections were stained using Hematoxylin and Eosin (H&E) to pinpoint the spatial location of each islet for downstream analysis (Fig. S7B).

MALDI-MS imaging of the mouse pancreas was followed by receiver operating characteristic (ROC) univariate analysis to identify $n = 345$ peaks differentially enriched in AL and CR islets or in islets versus acinar compartments (Fig. 5C and S7C, D). Next, we performed metabolite database query to identify ~50% of all ions detected (Supplementary Data 8 and 9). For example, acinar-enriched metabolites include the phospholipid cyclic phosphatidic acid (*m/z:* 391.22486, CPA 16:0) and nucleotide cytidine 2' 3'-cyclic phosphate (*m/z:* 304.03402, CMP), which is involved in the pyrimidine metabolism. In contrast, islet-enriched metabolites were largely classified as lipids, including cholesterol sulfate (m/z: 465.30443) and phospholipids (e.g., CPA 18:0 or different types of phosphatidic acids (PA)) (Fig. 5D).

Next, we applied a similar analysis to identify metabolites enriched in AL or CR islets, including $n = 28$ ions enriched in AL islets versus $n = 52$ ions enriched in CR islets (Fig. 5E, Supplementary Data 8). While AL islet-enriched metabolites were mostly lipids and/or signaling molecules, CR islet-enriched metabolites also included nucleotides and glucose metabolism metabolites such as uridine 2',3'-cyclic phosphate (UMP) and CMP, hexose 6-phosphate (glucose-6-phosphate or fructose-6-phosphate, m/z: 259.02236, G/F6P), cAMP (m/z: 328.04528), ADP (m/z: 408.0112), and cyclic ADP-ribose (m/z: 540.05334) (Fig. 5E, Supplementary Data 8, 9). These results indicate that CR associates with changes in islet lipid composition and increases the levels of molecules generally involved in beta cell homeostasis and insulin release pathways.

### CR reduces the expression of beta cell aging hallmarks

Aging beta cells have increased ER stress, defective autophagy, accumulation of DNA damage marker *53BP1*, and loss of the nuclear lamina protein Lamin B1 (*Lmnb1*)[12,18,48]. Together with insulin resistance, these markers are thought to contribute to beta cell senescence, islet inflammation, and impaired beta cell function during old age[18,19,49]. We hypothesized that the lower beta cell workload of CR mice would be associated with lower expression of beta cell aging markers. Therefore, we exposed adult male mice to CR for up to 12 months and found that aged CR mice phenocopied younger CR mice with lower body weight due to reduced fat content, and lower meal-stimulated beta cell insulin with increased peripheral insulin sensitivity (Fig. 6A–H and Fig. S8A). Of note, these mice also had smaller pancreases and lower islet beta cell mass, which suggested that CR could induce the arrest of normal organ growth via proliferation (Fig. S1G)[50,51].

Next, we used IHC to measure levels of *53BP1* and *Lmnb1* protein, and mRNA Fluorescent in Situ Hybridization (FISH) to quantify the expression of senescence-associated genes *p16/Cdkn2a* and *p21/Cdkn1a*. Remarkably, 12 months of CR reduced age-dependent accumulation of *53BP1*, sustained higher *Lmnb1* levels, and reduced the

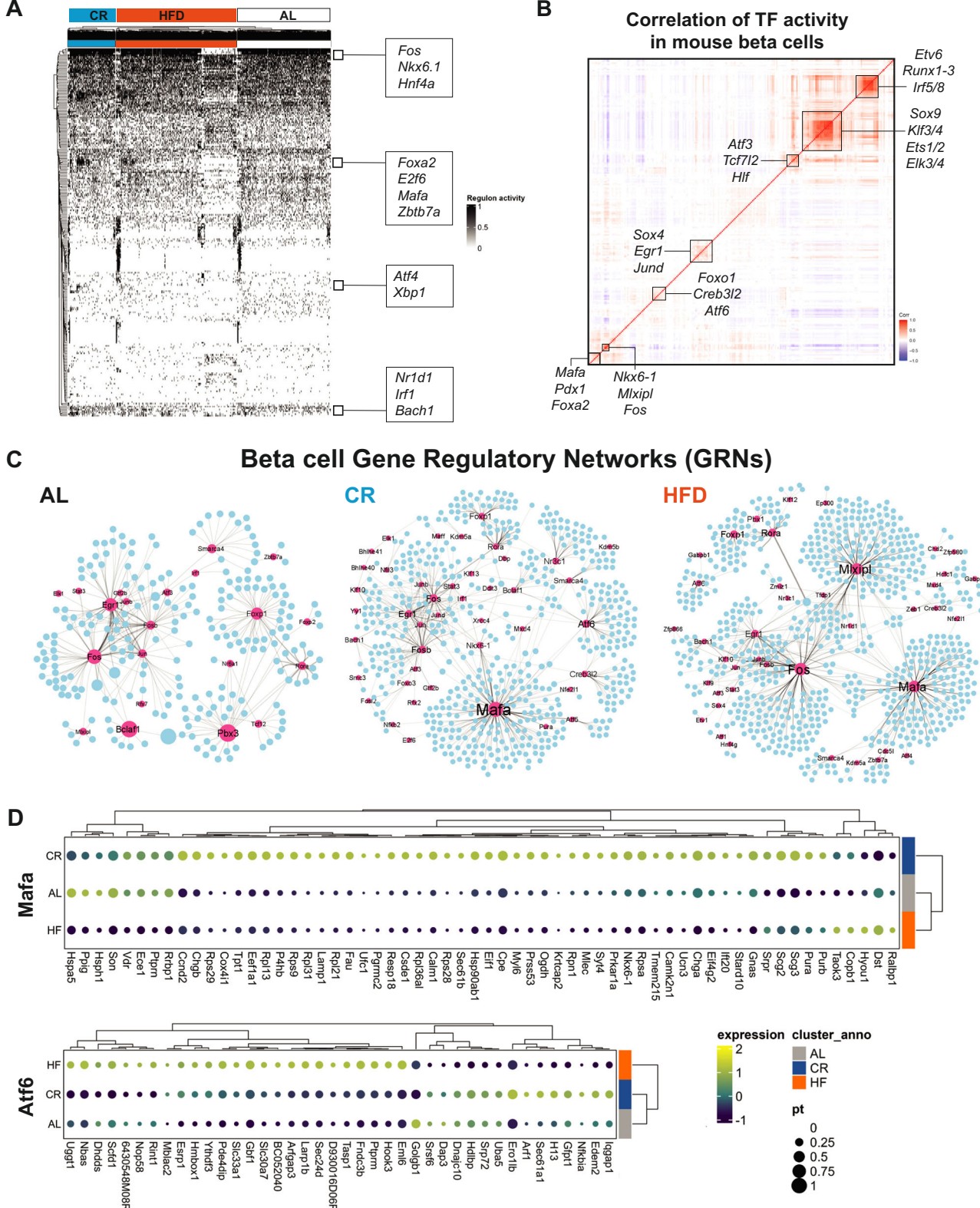

**Fig. 4 | CR reprograms beta-cell gene regulatory networks (GRNs). A** SCENIC heatmap with hierarchical clustering analysis of TF activity in beta cells from AL, CR and HFD mice after 2 months on diet. TFs identified as "ON" are shown in black, while TFs identified as "OFF" are in white. **B** Pearson correlation matrix of *n* = 272 TF identified in all mouse beta cells (all diet groups together). Boxes highlight clusters of TFs with a high degree of correlation. **C** Gene regulatory networks (GRNs) formed by TFs identified using SCENIC in beta cells from AL, CR and HFD mice. TFs are shown as pink nodes, while target protein-coding genes are shown in blue. Node size represents the "betweenness centrality" measurements that report the influence of a given TF within a network. **D** Dot plot and hierarchical clustering showing the gene expression levels of target genes associated with Atf6 and Mafa GRNs from AL, CR, and HFD beta cells. The dot plot scale shows the relative expression level and percentage of cells expressing a target gene.

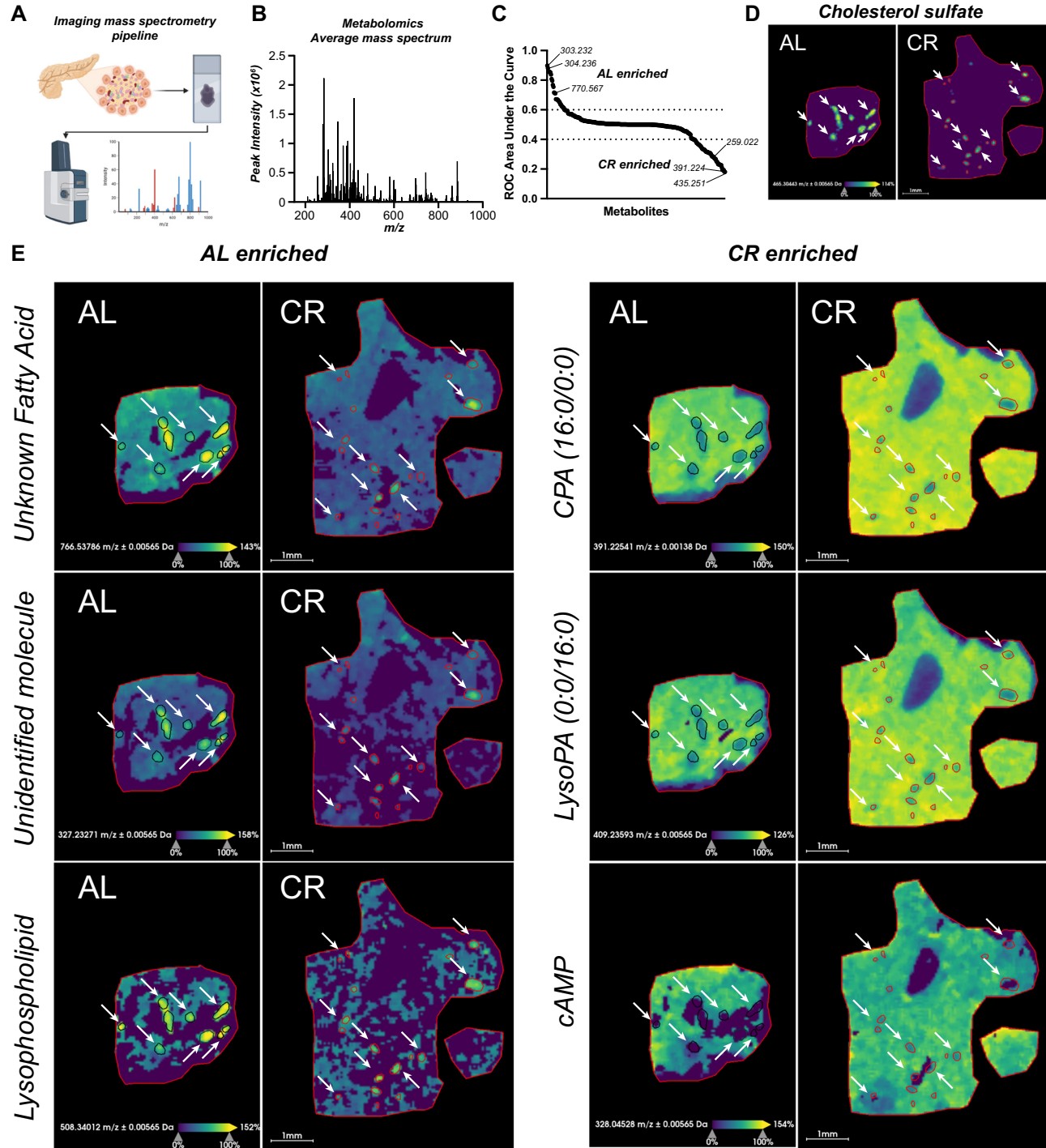

**Fig. 5 | CR beta cells are metabolically fit. A** Schematic diagram of imaging mass spectrometry (MALDI-MS) approach to measure metabolite abundance in AL and CR pancreases after 2 months on diet. **B** Average mass to power (*m/z*) spectra of all samples combined. **C** Receiver Operating Characteristic (ROC) analysis of ions enriched in AL versus CR islets. **D** Representative images of an islet-enriched ion (cholesterol sulfate, m/z: 465.304 Da). **E** Representative images of AL or CR islet-enriched ions. **C** Data pooled from *n* = 3 sections (1 section per mouse) per diet group. **D, E** scale bar = 1 mm. **A** was created with BioRender.com. and released under a Creative Commons Attribution-Non-Commercial NoDerivs 4.0 International license.

total expression of *p16/Cdkn2a* and *p21/Cdkn1a* in situ (Fig. 6I–K, Fig. S8C, D). Notably, most beta cells analyzed had detectable *p16* (96% AL vs 94% CR, *p* > 0.05) or *p21* (61% AL vs 59% CR, *p* > 0.05) puncta in nuclear and/or cytoplasmic compartments with no correlation in the expression of these two markers at the single cell level (Fig. S8D). Based on this data, we conclude that 12 months of sustained CR results in an insulin-sensitive state that lowers the demand for beta cell function and insulin release. In turn, this phenotype is associated with

reduced expression of 4 different markers of beta cell aging and/or senescence.

### CR enhances beta cell autophagy and mitochondria structure–function
Aging beta cells have impaired autophagic flux that may contribute to and/or explain the accumulation of ER stress and lipofuscin bodies[9,12]. Normalization of normal beta cell autophagy may assist in improving

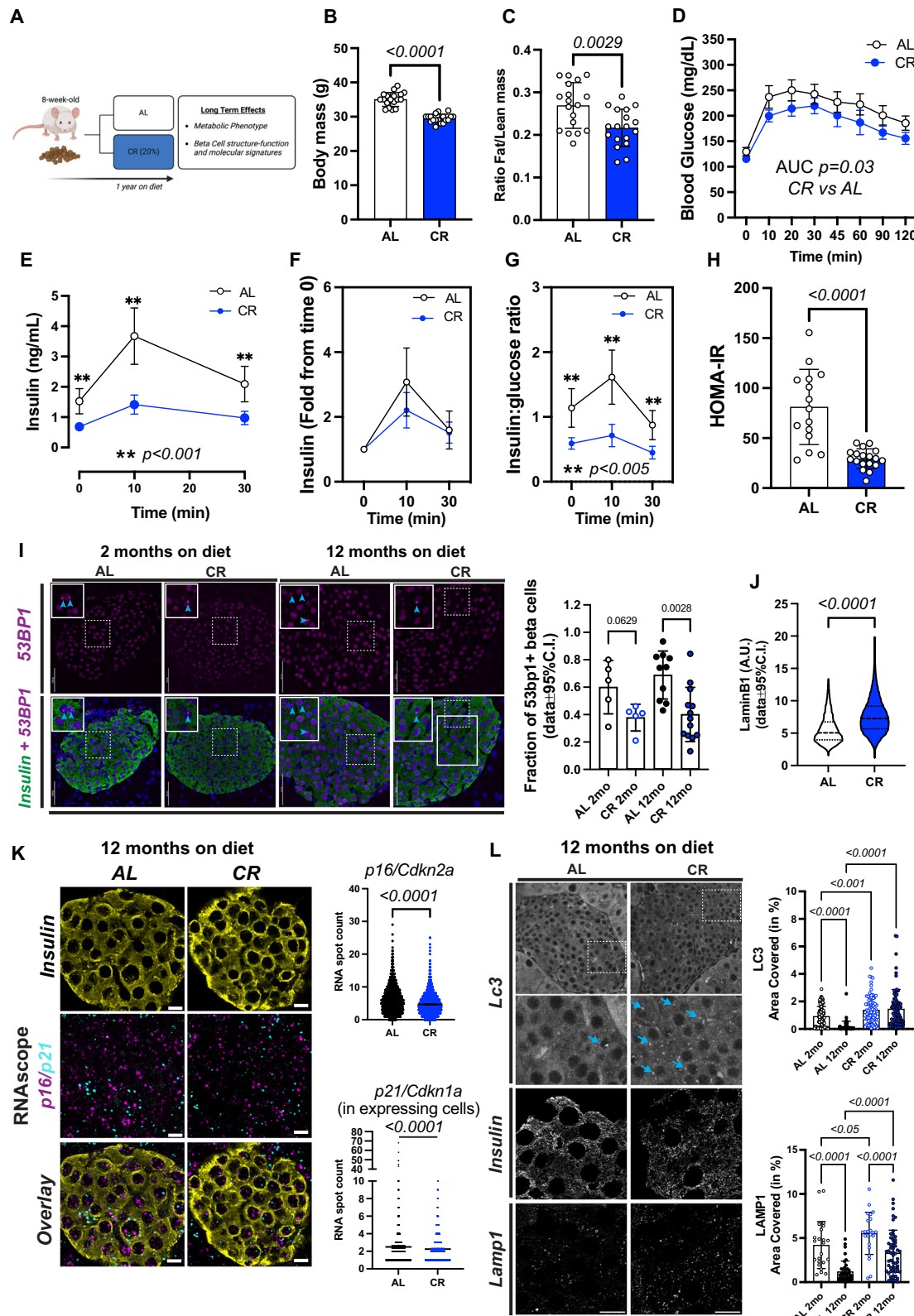

compromised beta cell function, as demonstrated by the activation of beta cell autophagy pathways via intermitted fasting to maintain normal beta cell mass and function in HFD mice[34]. We hypothesized that the enhanced adult beta cell phenotype found in young and old CR mice would be associated with increased autophagy. Indeed, our single-cell transcriptomics shows that CR beta cells have higher levels of autophagy genes (*Beclin-1, Lamp1*) and transcriptional regulators of

autophagy (*Tfeb, Tfe3*). We validated these findings and the impact of CR on beta cell autophagy during aging via confocal microscopy of CR islets stained with the autophagosome marker *Lc3I-II* and the lysosome marker *Lamp1*. As suggested by our omics data, CR significantly increased *Lc3I-II* and *Lamp1* vesicle density in CR beta cells after 2 or 12 months on diet (Fig. 6K, L, Fig. S8E), and prevented the loss of *Lc3I-II* and *Lamp1* positive vesicles during aging (Fig. 6K, L). These results are

**Fig. 6 | Long-term CR delays beta cell aging signatures. A** Schematic diagram of mice subjected to AL or 20% CR for 12 months starting at the age of 8 weeks. **B** Body mass after 12 months on diet. **C** Ratio between fat mass and lean mass after 12 months on diet. **D** Blood glucose levels during the meal tolerance test (MTT) after 12 months on diet. **E** Blood insulin levels during the MTT and **F** the respective fold change from baseline values. **G** Ratio between the insulin and glucose values obtained during the MTT. **H** HOMA-IR calculated from fasting glucose and insulin values after 12 months on diet. **I** Representative images of pancreatic sections from AL and CR male mice after 2 months or 12 months on diet stained with 53BP1 and insulin. Right, quantification of DNA damage by 53BP1+ beta cells. Each dot represents the average from each mouse. A total of 127 AL and 200 CR islets were analyzed. Data are normalized by percentage of 53BP1-positive beta cells per total beta-cell number. Scale bar, 50 microns. **J** Quantification of beta cell nuclear levels of Lamin B1 in pancreatic islets from AL and CR mice after 12 months on diet. A total of 74 AL and 105 CR islets were analyzed. **K** Representative images of pancreatic sections from AL and CR male mice after 12 months on diet stained with Cdkn2a and Cdkn1a mRNA probes, and insulin. Right, the quantification of incidence of these markers per beta-cell. A total of $n = 5570$ AL and $n = 4781$ CR beta cells were

analyzed. Scale bar, 5 microns. **L** Representative images of pancreatic sections from AL and CR mice after 12 months on diet stained with Lc3I-II or Lamp1. Right, quantification of beta cell area occupied by Lc3I-II or Lamp1. Each dot represents the average measurement from individual islets. A total of $n = 108$ AL 2mo, $n = 89$ CR 2 mo, $n = 51$ AL 12mo, and $n = 73$ CR 12mo islets were analyzed for Lc3I-II staining. A total of $n = 23$ AL 2mo, $n = 23$ CR 2 mo, $n = 38$ AL 12mo, and $n = 57$ CR 12mo islets were analyzed for Lamp1 staining. Scale bar, 10 microns. **B, C** $n = 18$ mice per diet group; **D–H** $n = 15$ AL and $n = 18$ CR mice per group; **I** $n = 5$ AL 2mo, $n = 5$ CR 2mo, $n = 10$ AL 12mo, and $n = 13$ CR 12mo mice per group; **J–L** Data pooled from $n = 5$ male mice per diet group. $P$ values for all significantly different comparison are shown. Statistical analysis was conducted using two-way ANOVA with Sidak's post-hoc test **E–G**, or one-way ANOVA with Benjamini, Krieger and Yekutieli's post-hoc test (**I, L**). For two groups comparison, unpaired two-tailed Student's $t$ test was performed (**B, C, H, J,** and **K**). For all panels, data presented as mean ± 95% confidence intervals (C.I.). **A** was created with BioRender.com. and released under a Creative Commons Attribution-Non-Commercial NoDerivs 4.0 International license.

---

further supported by downregulation of Ser 242/244 phosphorylation of the mTORC1 target ribosomal protein S6 (*p-rpS6*), which is required to activate autophagy[52] (Fig. S8F, G). Furthermore, this effect was specific to CR feeding since the intake of a calorie-diluted diet was unable to suppress *p-rpS6* and/or activate beta cell autophagy in DL mice, as evidenced by confocal microscopy of the autophagy marker *p62/Sqstm1* (Fig. S8H, I).

Autophagy activated during conditions of nutrient deprivation is important for cell survival by supplying metabolites generated from the degradation of organelles and/or macromolecular complexes via the lysosome system[53]. To understand what the potential targets of CR beta cell autophagy could be, we first performed deconvolution-assisted confocal colocalization microscopy of mouse islets stained for lysosomes (*Lamp 1*) and two large organelle "sources" in the beta cell cytoplasm: insulin granules (*Ins*) and mitochondria (marked using antibodies against succinate dehydrogenase isoform A (*Sdha*)). Surprisingly, we found no changes in the colocalization between *Ins* and *Lamp1*, which means that CR-induced autophagy does not target mature insulin granules (Fig. S9A)—which is in contrast to what occurs during short periods of fasting[54]. Next, we decided to measure the relative levels of the autophagy adaptor protein *p62/Sqstm1*, which targets poly-ubiquitinated (Ub) protein complexes to proteasome and lysosome machineries[55] and is increased in CR beta cells via single-cell transcriptomics (Fig. S9B). Here, confocal microscopy of *p62* and *Lamp1* revealed that CR increases beta cell *p62*-vesicle density without affecting *p62-Lamp1* colocalization (Fig. S9E). This suggests that CR promotes a proportional increase in the number of autophagic events and autophagic flux, which could potentially mediate the degradation of ubiquitinated proteins and explain the increased expression of several Ub processing, Ub protein degradation, and proteasome subunit genes (Fig. S5A, S9F).

These experiments also revealed that CR beta cells have lower levels of *Sdha-Lamp1* colocalization (Fig. 7A). We hypothesized that beta cell mitophagy would be decreased during CR and lead to higher beta cell mitochondrial content. This idea is initially supported by lower expression levels of the mitophagy adaptor Parkin (*Prkn*) and by higher expression of several mitochondrial-associated genes in these cells (Figs. 3 and 4, Fig. S9B). This phenotype appears to be specific to mitochondria since genes involved in the degradation of ER (called reticulophagy) are not differentially modulated by CR (Fig. S9D). To test our hypothesis, we applied high-resolution scanning electron microscopy (SEM) to quantify beta cell mitochondrial density in AL/CR islets in situ. This approach also allowed us to measure mature insulin granule content, and that was not different between AL and CR beta cells (as expected, Fig. S9C). Importantly, beta cell SEM revealed that CR beta cells indeed have higher mitochondrial mass (Fig. 7B, C).

Together, this data so far indicated that CR reduced beta cell mitophagy to increase mitochondrial density, whereas our single cell analysis quantified the higher expression of several mitochondrial function genes, including members of the electron transport chain (e.g., *Cox* genes, and the outer membrane pore complex *Vdac1*)) in CR beta cells (Figs. 7 and S9G). We hypothesized that this phenotype would be translated into higher mitochondria efficiency that would maintain an active beta cell metabolic homeostasis machinery (Figs. 3–5). To investigate beta cell mitochondria structure-function in situ, we applied EM tomography (eTomo) using AL and CR mouse pancreases from our 2-month cohort prepared for SEM (see methods for details, Fig. S10A)). eTomo provides high-resolution measurements of the mitochondrial structure and integration of anatomical measurements of mitochondria morphology, cristae number, cristae membrane surface area and density, and type of cristae structure (e.g., lamellar versus tubular) to estimate the functional capacity and potential of individual mitochondria to generate ATP[56]. We used 300 nanometer-thick beta cell sections and achieved an isotropic axial resolution of 1.62 nm, which allowed us to reconstruct the 3D architecture of individual beta cell mitochondria and mitochondrial cristae (Fig. 7D, E, Fig. S10A, Video S1). 3D segmentation of individual cristae was achieved using deep learning-based segmentation pipelines, as previously established by us[36]. This volumetric super-resolution EM imaging approach revealed that most beta-cell cristae are of the tubular type, and that CR increases beta-cell mitochondria cristae surface area and cristae density (Fig. 7E–G, Fig. S10B, C) without altering mitochondrial volume (Fig. S10C). Notably, the tubular morphology of beta cell mitochondria (Fig. S10D) can be regulated by *Opa1* and MICOS complexes[57], and which are up regulated in CR beta cells (except for *Apool* and *Chchd3*, Fig. 7H). Finally, we calculated the potential rate of ATP molecules generated per second per mitochondrial volume using an established biophysical mathematical model[58], and we estimate that each CR beta cell mitochondria can produce up to ~59,000 ATP molecules/second/mitochondrial volume; 14% higher than in AL beta cells (Fig. 7I).

While analyzing the eTomo volumes, we noticed vesicle-like structures stemming from the outer membrane regions of beta cell mitochondria that resembled mitochondrial-derived vesicles (MDVs, Figs. 7D, J, and Fig. S10E–G). MDVs are part of mitochondrial proteome quality control pathways that deliver oxidized/damaged macromolecular complexes to the lysosomal degradation system, and/or that can act as vehicles of inter-organelle transport[59,60]. We found MDVs in ~50% of AL beta cell mitochondria, whereas MDVs in CR beta were very rare (Fig. 7J). We speculate that this could be due to higher expression of oxidative molecular scavenger proteins Superoxide Dismutase 1 and 2 (*Sod1/2*, Fig. S9H), which could lead to reduced

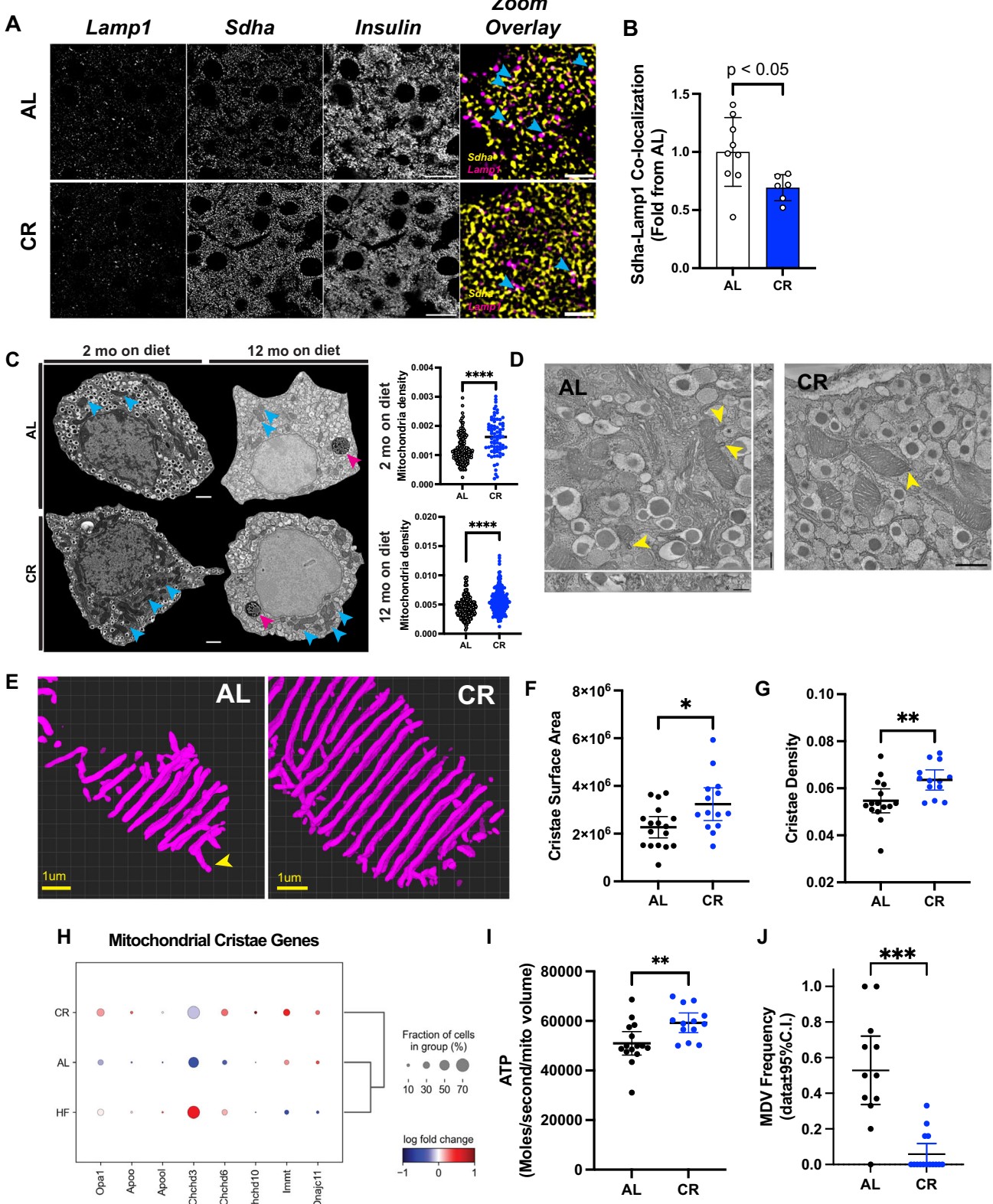

accumulation and degradation of damaged proteins. Accordingly, most beta cell MDVs contained membrane-rich cargo that fused with lysosomes (Fig. S10E, F), and only one example was found fusing with the ER. Therefore, we propose that beta cell MDVs are mostly implicated in mitochondrial protein quality control by delivering cargo to the lysosome machinery (Fig. S10G).

These results show that CR increases beta-cell autophagy, likely to promote the turnover of macromolecules via the Ub and lysosome systems, whereas beta-cell mitochondrial structure is protected by reduced rates of mitophagy and CR-dependent increases in beta cell mitochondria cristae density are expected to facilitate increased ATP production. This is associated with reduced formation of beta cell

**Fig. 7 | CR increases beta cell mitochondria density and modifies cristae structure. A** Representative images of pancreatic sections from AL and CR male mice after 2 months on diet. Slides were stained with Insulin, Lamp1, and Sdha. Blue arrowheads point to regions of overlap between Sdha and Lamp1 signal. **B** Colocalization between Sdha and Lamp1 was measured. Each dot represents the average colocalization index in each animal calculated from ~100 beta cells per animal. **C** Representative images of pancreatic beta cells from AL and CR male mice using electronic microscopy. Blue arrows point to mitochondria density, while pink arrows point to lipofuscin granules in aged beta cells. Mitochondrial density is measured by the total mitochondrial number per beta cell area. Each dot represents a single beta cell analyzed from three different islets per mouse (n = 3 mice per diet group). **D** Representative images of pancreatic beta cells from AL and CR male mice using high-resolution electron tomography (eTomo). Yellow arrowheads indicate MDVs, whereas the asterisk in each image marks the location of lysosomes.

**E** Representative 3D reconstructions of beta cell crista segmentation generated using deep learning image analysis tools. **F, G** Cristae surface area and cristae density in AL or CR beta cells. **H** Dot plot showing the relative expression levels of genes involved in cristae formation and morphology. **I** Calculation of ATP generation for an average beta cell mitochondrion. Each dot represents data from one eTomo image stack. **J** Relative frequency of mitochondrial-derived vesicles observed in eTomo images. **F–H** and **I, J** each dot represents an individual field of view with an average of five mitochondria per field. eTomo data acquired from n = 3 mice per diet group, five fields of view per animal. Total of 69–70 mitochondria analyzed per diet group. The asterisks indicate *$p < 0.05$, **$p < 0.01$, ***$p < 0.001$, and ****$p < 0.0001$ using unpaired, two-tailed Student's t test. **C, D** Scale bar = 500 nm. **A** Full view panels have a scale bar = 10 microns. Zoom overlay scale bar = 2 microns. **E** scale bar = 1 micron. All data presented as mean ± 95% confidence intervals (C.I.).

MDVs, which is indicative of reduced turnover of mitochondrial proteome turnover and/or organelle inter-communication.

### CR promotes beta cell longevity

The reduced pancreas and islet beta cell mass of CR mice suggested that CR is associated with a mostly post-mitotic cell state with reduced expression of beta cell aging markers and enhanced cell and organelle homeostasis (Figs. 1–7). We hypothesized that these molecular and structural signatures of CR beta cells indicated that beta cell turnover was reduced, which would promote a more mature and LLC phenotype. Here, to quantify beta cell turnover and longevity in situ, we applied stable isotope labeling of mice using $^{15}$N-enriched diet followed by a correlated SEM and multi-isotope mass spectrometry (MIMS) microscopy approach (called MIMS-EM) developed previously by us to quantify cell and protein complex age[4,7]. Here, Nitrogen 15 ($^{15}$N)-labeled animals were created *in utero* and maintained on $^{15}$N-diet until post-natal day 45 (P45). Next, $^{15}$N-mice were randomly allocated to AL, CR, or HFD feeding groups using $^{14}$N-rich food pellets and maintained on their respective diets for 12 months (i.e., the chase period). Finally, the mice were prepared for MIMS-EM imaging and quantification of cell age based on the amount of $^{15}$N retained in the cell nucleus (Fig. 8A, B)[4].

To quantify the maximum amount of $^{15}$N incorporated by P45 as well as the $^{15}$N levels remaining in the nucleus of post-mitotic cells after a 1-year lifespan on $^{14}$N-diet, we performed MIMS-EM of brain neurons after 6 or 12 months of chase. This approach revealed that P45 mice were saturated with $^{15}$N and, after 6-12 months of chase, most of the $^{15}$N signal was restricted to the cell nucleus and myelin structures (Fig. 8C and Fig. S11A, B), as expected[6,12]. Next, we applied MIMS-EM to islet cells from AL, HFD, or CR mice and found that beta cells at P45 were fully saturated with $^{15}$N, just like neurons (Fig. S11C). Importantly, the levels of nuclear $^{15}$N in islet and acinar cells were heterogeneous, which is indicative of distinct populations of beta cells with different longevities (Fig. 8D)[4]. We found that HFD beta cells had significantly lower $^{15}$N levels than AL mice (Fig. 8D, E), while CR beta cells had higher $^{15}$N levels (Fig. 8D, E). No effects of HFD or CR on alpha or acinar cells were observed; however, delta cells in CR islets had higher $^{15}$N levels (Fig. S11D). We then classified each beta cell as a potential LLC or a relatively younger cell according to their nuclear $^{15}$N content using cortical neurons as a reference[4], and found that up to 80% of all beta cells in CR mice are LLCs (Fig. 8F). Surprisingly, ~30% of all HFD beta cells were LLCs and suggests that a small population of beta cells may be resistant to HFD-induced proliferation (Fig. 8F).

### Discussion

In this study, we investigated how CR impacts adult beta cell insulin release, metabolism, transcriptional heterogeneity, and aging signatures. We exposed adult male and female mice to a 20% reduction in daily caloric intake by providing a limited amount of food before the start of their active period at night. Recent studies have shown that

conventional CR achieved via TRF is associated with long fasting periods that are important for the beneficial metabolic and geroprotective effects of CR[29,61–63]. These effects are sex-dependent due to differences in glucose homeostasis and metabolism[21,63–65]; importantly, our results further support the link between increased insulin sensitivity and/or enhanced insulin signaling to increased cellular longevity promoted by CR and CR-mimicking approaches[66–68]. Due to its beneficial effects on glucose homeostasis and body fat content, CR has been proposed as a feasible approach to combat cardio-metabolic diseases like T2D by reducing the metabolic demand for insulin release and by lowering blood glucose levels. Our data indicates that maintenance of an insulin-sensitive state achieved via CR lowers the metabolic demand on beta cells to secrete insulin and maintain normoglycemia (Fig. 1). The link between insulin sensitivity and beta cell function was further reinforced in mice fed a diluted calorie diet, which was insufficient to improve insulin sensitivity and modulate beta cell function. These results add support for the long-proposed idea of beta cell "rest" as a potential strategy to recover (at least temporarily) failing beta cell function during T1D and T2D[69,70]. Strikingly, CR beta cells fail to engage in a normal adaptive proliferation response when exposed to HFD, potentially due to changes in their overall transcriptional architecture, which creates a scenario where CR beta cells achieve higher insulin release levels with ~30% less beta cells (Fig. 2). More studies are needed to determine whether longer exposures to HFD would lead to CR beta cell failure.

How do adult beta cells adapt to CR? To answer this question and study the underlying molecular landscape of CR beta cells, we took a multiomics approach. First, using single nucleus transcriptomics to measure beta cell gene expression and chromatin accessibility, together with TF activity inference tools, we show that CR beta cells activate a transcriptional response associated with enhanced beta cell structure-function and identity achieved via transcriptional modulators of circadian rhythm (*Dbp*, *Bmal*), beta cell identity (*Mafa*, *Nkx6.1*), and regulation of ER stress, cell homeostasis, and autophagy (*Atf6*, *Tfeb*) (Figs. 3 and 4). We validated these results in situ using light, electron, and mass spectrometry microscopy, including the higher levels of the circadian and beta cell identity transcriptional (*Bmal* and *Pdx1*, respectively), and the increased density of autophagy vesicles (*p62/Sqstm1*, *Lc3l-II*, and *Lamp1*) in CR beta cells (Figs. 3–4, 6). This data is in line with previous studies where CR and TRF restored previously compromised beta cell function by rescuing gap-junction connectivity, autophagic fluxes, and circadian regulators[30,33,34]. Moreover, our data indicates that activation of these pathways may depend on (at least in part) enhanced insulin sensitivity and not on calorie dilution alone.

In the beta cell, autophagy has been implicated in the degradation of insulin granules and/or mitochondria to protect beta cells from accumulating and/or damaged molecules and organelles at times of acute fasting, metabolic stress, or inflammation[12,34,54,71–73]. In contrast, our data indicates that CR-induced beta cell autophagy does not target insulin granules or the mitochondria. Instead, autophagy is likely

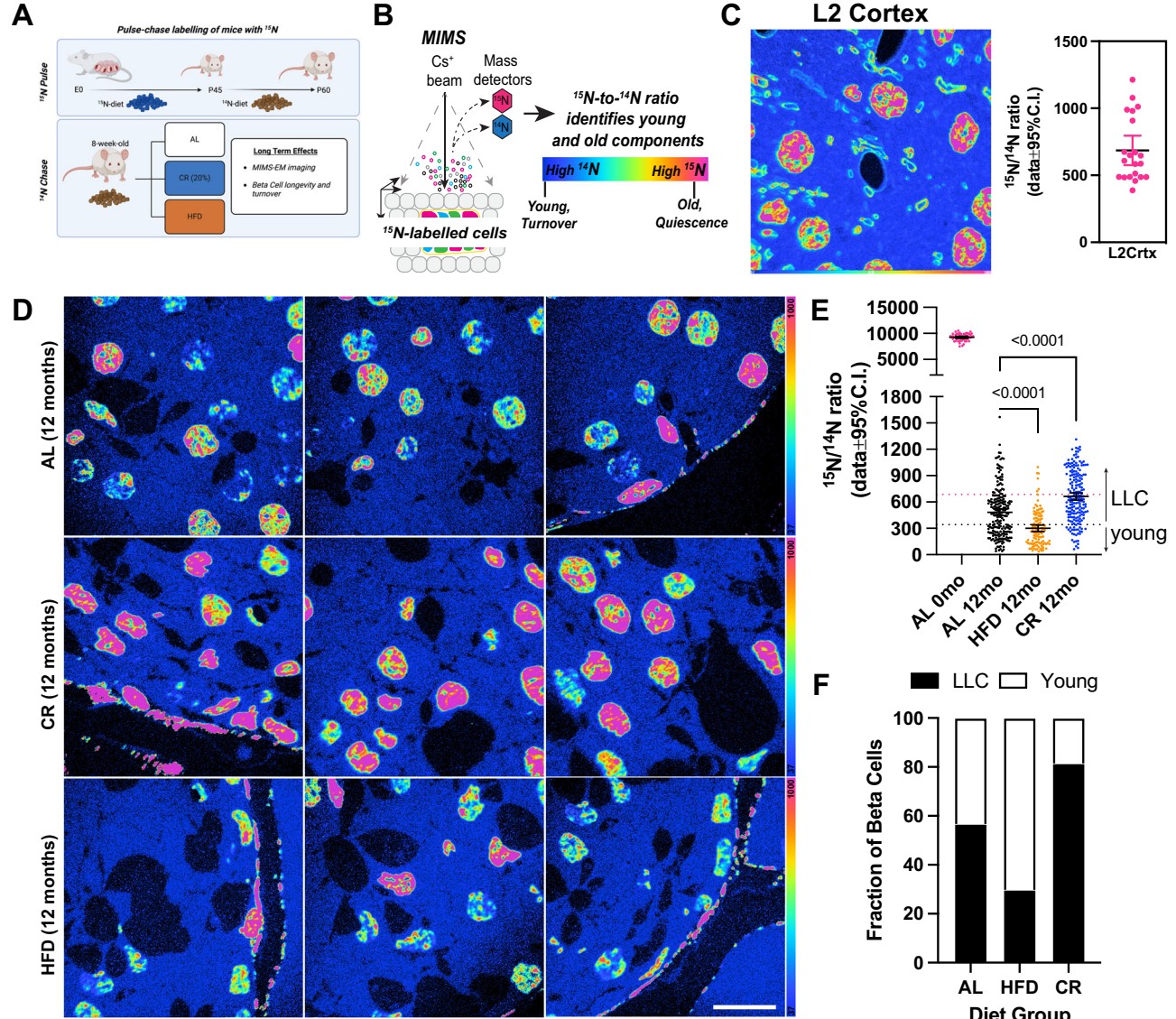

**Fig. 8 | CR promotes beta-cell longevity by slowing down beta-cell turnover rates. A** Study design. After 2 months of age, [15]N-labeled mice were kept on AL, 20% CR or HFD for 12 months. **B** Multi-isotope imaging mass spectroscopy (MIMS) representation technique. **C** Validation of [15]N incorporation by MIMS imaging and quantification of [15]N/[14]N ratio in cortical neurons from a [15]N-labeled mouse after chase with [14]N diet for 12 months. Each dot represents a single neuronal nucleus (*n* = 21 nucleus from 2 cortical layers of 1 mouse) **D** MIMS of pancreatic islets from AL, CR, or HFD mice (*n* = 3 islets per animal, *n* = 1 mouse per diet group). **E** Single cell analysis of [15]N/[14]N ratios in beta cells at day 0 (no chase) and after 12 months (mo) on AL, HFD, or CR diet. Each dot represents a single beta cell nucleus, pooled from 3 different islets of 1 mouse per diet group. Total of *n* = 54 AL

0mo, *n* = 217 AL 12mo, *n* = 114 HFD 12m0, and *n* = 194 CR 12mo beta cells were analyzed. The dotted horizontal pink and black lines represent the mean and the lowest [15]N/[14]N levels found in cortical neurons shown in (**C**). **F** Estimation of beta cells that are estimated to be LLC or young after 12 months on AL, CR, or HFD. **B**, **D** scale bar = 10 microns. The *p* values for all significantly different comparison are shown. Statistical analysis was conducted using one-way ANOVA with Benjamini, Krieger, and Yekutieli's post-hoc test (**B**). All data presented as mean ± 95% confidence intervals (C.I.). **A** was created with BioRender.com. and released under a Creative Commons Attribution-Non-Commercial NoDerivs 4.0 International license.

involved in the degradation of poly-Ub substrates delivered to proteasomes and/or lysosomes via *p62*-positive autophagy vesicles. In addition, we show that CR beta cells protect their mitochondrial mass by reducing mitophagy, which explains why these cells have higher mitochondrial density. CR beta cells also have elevated expression of genes associated with electron transport chain components, cellular respiration, and cristae morphology (Fig. 3). Together, these signatures indicated that beta cell mitochondrial metabolism was increased by CR. We support this hypothesis with three separate lines of observations: First, MALDI-MS metabolomics reveals elevated levels of lipids and signaling molecules, cAMP, glucose metabolites, and ADP-ribose in CR beta cells – the later which could explain why CR beta cells

have lower DNA damage marker expression[74] (Fig. 5). Importantly, activation of beta cell metabolism and signaling pathways (including cAMP) is regulated by TFs enhanced by TRF (e.g., *Dbp*)[33] and CR (Figs. 3 and 4). Second, using eTomo microscopy, we show that CR increases mitochondrial cristae density, which translates into a higher potential for ATP synthesis (Fig. 7). Third, the near absence of MDVs in CR beta cells together with the upregulation of superoxide scavengers *Sod1* and *Sod2* strongly suggests that reactive oxygen species (ROS) generation and degradation of mitochondrial proteins is reduced during CR. Accordingly, CR increases mitochondrial bioenergetic efficiency and reduces oxidative stress accumulation in several systems[21,75,76], which may increase mitochondrial protein longevity in

aged cells[77]. Reduced accumulation of ROS and enhanced bioenergetics could also explain why CR beta cells secrete less insulin, as dysregulation of ROS and glycolytic signaling are proposed as causes of basal hyperinsulinemia during diabetes[78,79]. We speculate that higher ATP production of CR beta cells could serve other ATP-dependent processes (e.g., ER calcium homeostasis, protein folding, or proteasome function) instead of K-ATP channel closure and insulin release. Future experiments are needed to measure ER and mitochondrial homeostasis to further understand how CR regulates insulin secretion during basal and nutrient-stimulated conditions.

Finally, using $^{15}$N-SILAM followed by MIMS-EM we quantify beta cell age in situ in mice exposed to CR or HFD to demonstrate that CR prolongs beta cell longevity by reducing cell turnover. Given that the share of long-lived beta cells decreases during HFD, we speculate that these cells could act as a "beta cell reserve pool" that participates in the adaptive beta cell proliferation response at times of metabolic stress (Fig. 8). However, long-lived beta cells are still found in the HFD islet, thus suggesting that some beta cells do not proliferate even after 1 year on HFD. Previous studies have established that proliferation promotes an immature and less functional beta cell phenotype than non-proliferating beta cells[80]. In contrast, CR beta cells are largely postmitotic and express higher levels of beta cell identity and function genes (Figs. 3 and 4). Furthermore, we show that this largely postmitotic beta cell state during CR associates with downregulation of several beta cell aging phenotypes, including the rescue of autophagy homeostasis and reduced expression of *LamnB1*, *p16/Cdkn2a*, and *p21/Cdkn1c* (Fig. 6). A reduction in beta cell turnover may also explain the reduced accumulation of *53BP1*, as cell proliferation triggers higher accumulation of genetic mutations and replication-dependent stress linked with aging[81].

In conclusion, our study shows that a mild CR approach enhances insulin sensitivity to promote beta cell health and longevity. This is achieved via the upregulation of several beta cell identity and energy homeostasis mechanisms while lowering the expression of beta cell ageing markers. These results offer a molecular footprint of how CR affects adult beta cells and suggests that CR could preserve and/or rescue beta cell dysfunction during aging.

## Limitations of our study

The results reported here determine how CR impacts mouse beta cell heterogeneity in vivo and in situ. Here, CR was efficiently achieved by providing a restricted amount of food before the start of the animal's active phase. By design, this approach potentializes the metabolic impacts of CR (including those promoted by prolonged fasting periods) mediated by activation of circadian rhythm mechanisms and nutrient metabolism[29,64]. To dissociate the contribution of CR from enhanced insulin sensitivity, we performed experiments where mice were fed "diluted" food pellets. While this approach leads to a reduction of daily caloric intake without affecting insulin sensitivity (this study and[29]), DL-fed mice adapt to consume enough calories to support normal energy levels over time, which may be due to changes in metabolism and feeding behaviors[82]. Therefore, additional experiments are needed to further dissect the degree to which enhanced insulin sensitivity contributes to the known and beneficial effects of CR. The $^{15}$N-SILAM and MIMS-EM experiments add to previous observations of cellular age mosaicism in somatic tissues, including in the pancreas[4,7,83]. Like MALDI-MS metabolomics and eTomo, this approach is time-consuming and financially expensive (thus limiting the number of animals and cells analyzed). Notably, the nuclear $^{15}$N measurements of any cell can be variable due to heterochromatin-rich domains present at different frequencies in the sections analyzed. These compartments tend to have a higher density of $^{15}$N-DNA with higher signal intensities versus DNA "poor" euchromatin domains[4]. Nevertheless, our results confirm previous MIMS-EM results[4], and now show how beta cell longevity is modulated by dietary interventions.

## Methods

### Study approval
All animal procedures were approved by the Institutional Animal Care and Use Committee (IACUC) of Vanderbilt University (M2000086-00/01).

### Animals and experimental design
Male and female FVB/NJ and C57BL/6 J mice were obtained from the Jackson Laboratory (Bar Harbor, ME) and then were housed in cages of five mice and maintained on a 12-hour light/dark cycle with controlled temperature (21 °C) and humidity. The mice had AL access to chow food (Rodent Diet 20 5053, PicoLab®) and water. At 2 months of age, the mice were randomized to AL diet, 20% caloric restriction (CR), or 60% HF for 2 or 12 months. Both diets were purchased from Lab Diet (CR and AL: Rodent Diet 5053, 4.11 Kcal/g; HF: Rodent Diet 58Y1, 5.10 Kcal/g; St. Louis, MO). For experiments involving mice previously exposed to AL or CR and exposed to AL (in the case of CR mice) or HFD, the diet was provided AL. The diet supplemented with 20% cellulose was custom-made using diet #5053 as a base and purchased from Inotiv (West Lafayette, IN). Animals exposed to this diet were fed AL. For the CR protocol, mice were fed with 80% of the daily consumption of the AL group, which corresponds to a moderate caloric restriction of 20% of the control group (AL) energy intake. CR mice were fed once a day between 4:00–5:00 pm. A gradual reduction in food intake was performed (by 10% to 20% per week) to avoid an abrupt decline in body weight and maladaptation. For all groups, mouse weight and food consumption were monitored daily. In vivo or ex-vivo experiments were performed after 2 or 12 months on diet, and the mice were euthanized by inhaled isoflurane. The experiments were performed in different cohorts of mice to confirm the phenotype and verify the robustness of our data. The number of mice used in each experiment is detailed in the figure legends.

### Body mass, food intake, and bomb calorimetry measurements
Body mass and food intake were checked daily in the first two months of diet and then weekly via digital electronic scale (Ohaus Compass, Parsippany, NJ). The body mass change was expressed as percentage changes relative to initial body mass[84]. For the body composition analysis, the fat mass and lean mass were measured on conscious mice after 2-or-12 months on diet by magnetic resonance using the Minispec-LF50 (Bruker Optics Inc, Billerica, MA). The food intake was determined by the 24 h difference between the food offered to the mice and the remaining food once per day (4:00–5:00 pm), divided by the number of mice in the cage. To calculate the energy available from digested food, we performed bomb calorimetry of fresh fecal samples collected within a 20-hour time span. After fecal samples were collected, they were shipped to the University of Michigan Animal Phenotyping Core for energy assessment. Samples were weighted, dried at 50 °C overnight, and weighted once more before being ground and stored at −80 °C. Finally, samples were measured using bomb calorimetry. The amount of digested energy from food (in kcal/day) was calculated by subtracting the energy in the total fecal mass from the total energy from the animal food intake.

### Glucose homeostasis assays
The MTT was performed after 2-or-12 months on diet in 6-h fasted mice (7:00 am–1:00 pm Central Time) without the use of anesthesia. The blood was collected by tail snip, and the blood glucose was measured before and after the meal bolus. The mice received a gavage of Commercial Ensure Original Shake (12 Kcal/Kg of body mass), and blood samples were collected from the tail tip before (minute −10) and after 5, 10, 15, 20, 30, 45, 60, 90 and 120 min of the meal administration for blood glucose measurements. At timepoints 0, 10 and 30, additional blood was collected for determination of plasma insulin levels. For experiments testing the effect of the GLP1R-agonist Liraglutide, mice

were pre-treated with 400 µg/kg body weight, 2 h before the start of the MTT experiment. Area under-curve of the MTT (AUC) was obtained after normalization by the initial blood glucose values. To measure insulin sensitivity, an intraperitoneal insulin tolerance test (ipITT) was performed after 2-or-12 months on diet in overnight fasted mice (9:00 pm–7:00 am Central Time) without the use of anesthesia. The blood was collected before (minute 0) and after 10, 15, 20, 25, 30, 40 and 60 min of insulin (Humulin regular U-100, Lilly, Indianapolis, IN) administration (1 U/Kg of body mass), as described for the MTT, for blood glucose measurements. The constant rate of glucose disappearance (Kitt) was calculated from the slope of the linear regression obtained with log-transformed glucose values between 0 and 60 min of the insulin loading[84]. The HOMA-IR was calculated from fasting glucose (mmol/l) and fasting insulin (mU/l) values obtained during the MTT (at timepoint -10).

### Insulin degrading enzyme (IDE) activity

The assay was performed in liver samples from mice after 2 months on diet by SensoLyte® 520 IDE Activity Assay kit (AnaSpec, Fremont, CA), according to the manufacturer's instructions. The kinetic concentration of 5-FAM and total IDE activity were normalized per µg of total protein as previously described[85].

### Islet isolation

Pancreatic islets were isolated from mice after 2-or-12 months on diet by collagenase digestion method. Briefly, a collagenase solution (0.6 mg/ml) was injected into the pancreatic duct and the isolated pancreas was digested at 37 °C for 10 min. Cold Hank's Solution (HBSS) with 10% fetal bovine serum (FBS) was added to stop the enzyme activity. After centrifugation the pellet was washed with HBSS 10% FVB and then the islets were purified with a Histopaque gradient. After a new set of washing and centrifugation the islets were hand-picked. The fresh islets were used for perfusion assay or immediately snap freeze for RNA/DNA extraction.

### Islet perfusion and Insulin secretion

Islet perfusion was performed in fresh islets from 2 months on diet mice as described in ref. 86. Dynamic insulin secretion was assessed using a cell perifusion system with islets matched for size and number (IEQ). Islets from each mouse were preincubated for 2 h and then perifused with 5.6 mM glucose (9 min), 16.7 mM glucose (30 min), 5.6 mM glucose (21 min), 16.7 mM glucose + 100 µmol/l iso-butylmethylxanthine (IBMX) (9 min), 5.6 mmol/l glucose (21 min), 5.6 mM glucose + 20 mM KCl (9 min) and 5.6 mmol/l glucose (21 min). Three-minute fractions were collected for measurement of insulin by ELISA (Mouse Insulin ELISA, Mercodia, Sweden) while the insulin total content was determined using the islets extracts.

### Tissue processing and IHC

Tissues for IHC were processed as previously described[12]. Pancreas, liver, and adipose tissue from AL, CR, and HFD mice were dissected, weighed, and fixed in 10% formalin for 72 h at room temperature. Tissues were dehydrated, embedded in paraffin, and sectioned in 5µm-thick sections. Next, the tissue sections were pre-heated to 60 °C for 20 min before immersion in fresh xylene (Fisher Scientific, USA) for 40 min. After the paraffin removal, sections were rehydrated in a series of graded alcohols until DiH$_2$0 and then subjected to heat-induced antigen retrieval in fresh sodium citrate buffer (10 mM sodium citrate, pH 6) using a conventional microwave oven (20 min). Next, the slides were cooled down to room temperature and incubated for 1 hour with blocking and permeabilizing solution (0.3% Triton X-100, 3% bovine serum albumin (BSA; Sigma-Aldrich) in phosphate buffered solution (PBS)). After blocking, the tissue sections were incubated overnight at room temperature with primary antibodies. The following antibodies were used: guinea pig anti-insulin (Dako, #IR002) at 1:4 dilution, rabbit

anti-glucagon (Cell Signaling Technology, #2760) at 1:100 dilution, rat anti-somatostatin (Millipore Sigma, #MAB354) at 1:100 dilution, rabbit anti-53bp1 (Bethyl Laboratories, #IHC-0001) at 1:100 dilution, goat anti-PDX1 (gift from Christopher Wright, Vanderbilt University) at 1:10,000 dilution, rabbit anti-LC3A/B (Cell Signaling Technology, #12741) and Lamin B1 (E6M5T) (Cell Signaling Technology, #17416S) at 1:100 dilution, rat anti-LAMP1 (Developmental Studies Hybridoma Bank, #1D4B) at 1:50 dilution, and mouse anti-p62 from R&D systems (cat# MAB8028-SP) at 1:100 dilution. For experiments involving antibodies hosted in mice, we added a mouse-on-mouse blocking solution (M.O.M., cat# MKB-2213-1 from Vector Laboratories, 1:40 dilution) to our blocking/permeabilization buffer. The next day, the slides were rinsed five times in TBS for 10 min each wash and incubated for 1 hour at room temperature with 4',6-diamidino-2-phenylindole (DAPI, ThermoFisher, #62248) at 1:400 dilution and secondary antibodies (1:400, Alexa Fluor 488, 555, 594, and/or 647). Finally, stained slides were rinsed four times using PBS (10 min each wash) and then mounted in VectaShield (Invitrogen, #H-1000) prior to confocal imaging. Microscopy of all slides from matched experiments was imaged in consecutively.

### Confocal microscopy

Immunostained FFPE mouse pancreas slides were imaged using a Leica Microsystems Stellaris X5 confocal microscope fitted with a 20 × /0.75 numerical aperture (NA) multi-immersion objective (at 1024 × 1024 pixels with a ~ 270 nm pixel size and 300-500 nm Z-steps), or a 63 × /1.3 NA glycerol objective, and a white-light laser as previously described by us[12]. The 63x was used for imaging of autophagy and mitochondrial markers, where acquisition of the 3D image stacks was configured according to Nyquist acquisition parameters predetermined using the "Lightning"-confocal mode. Here, images had an X-Y pixel size of 50-80 nm and 0.148 µm in Z. After acquisition, confocal images were deconvolved in the LASX software using the adaptive mode. For quantification of relative fluorescence levels, we acquired photon counts using photon–arrival time gated photon-counting HyD2 detectors using following excitation (Ex) and emission (Em) bandwidths during data acquisition: DAPI: Ex: 405 nm, Em: 430 to 500 nm; Alexa Fluor 488: Ex: 499 nm, Em: 504 to 556 nm; Alexa Fluor 555: Ex: 553 nm, Em: 558 to 595 nm; Alexa Fluor 594: Ex: 590 nm, Em: 600 to 750 nm; and Alexa Fluor 647: Ex: 631 nm, Em: 640 to 750 nm.

### Analysis of confocal imaging data

Confocal stacks were processed using ImageJ/FIJI, and maximum projection images from 4 to 10 optical sections (~4–8 microns) were generated. Images were then loaded onto QuPath for nuclei segmentation and cell border expansion, and beta cells were automatically identified via quantification of relative mean insulin intensity levels inside each detected cell followed by thresholding to exclude non-beta cells. The DAPI channel was used as an input for nuclei segmentation. We used a 3-to-5-µm border extension parameter to estimate the location of the cell cytosol. We only analyzed data from beta cells that were included in this classification. Double-blind manual analysis of beta cell 53BP1 staining was performed to quantify the frequency of 53bp1 puncta in individual beta cell nucleus. Only beta cells with clear 53BP1-positive foci were counted as "positive" cells. For analysis of beta autophagy data, image stacks were processed to generate maximum projections representing 2 microns of the original stack. Detection of p62, Lc3- and Lamp1-positive granules as well as segmentation of beta cells was achieved by creating 2D U-nets to detect "spots" or insulin-positive regions in our imaging data (Aivia software (v11), Leica Microsystems). This generated binary segmentation masks delineating insulin-, p62-, LC3-, or Lamp1-positive regions in each image. Next, we used our established CellProfiler pipeline to quantify the area of the beta cell cytosol occupied by p62, LC3, or Lamp1 granules. Colocalization of Lc3 and Lamp1- was calculated using Imaris software (v9.9,

Oxford Instruments), as previously described by us[12]. For colocalization studies involving insulin, *Lamp1*, *Sdha*, and/or *p62*, deconvolved images were analyzed using ImageJ/Fiji program Coloc2[87]. For all our colocalization indexes, we used Mander's index after automatic threshold (tM1/2).

## Islet morphology

To analyze the endocrine pancreas morphology and beta cell mass, we selected two interspaced sections (5μm-thick and 250 μm apart from each other). After the IHC for insulin, glucagon, and somatostatin, as described before, the entire sections were scanned and QuPath and ImageJ software were used for image analysis. The relative mass of beta-cells, alpha-cells, and delta-cells was calculated by dividing their respective total immunostained areas by the total pancreatic section area and then multiplying by the total pancreas mass (mg). The proportion of beta-cells, alpha-cells, and delta-cells per islets was calculated by dividing the total area of the cells immunostained for insulin, glucagon, or somatostatin, respectively, by the total area of the islet, and expressed as %, as previously described[84].

## Islet cell nuclei isolation and single nucleus multiomics

We performed simultaneous profile gene expression and open chromatin sequencing from same single islet cells using the Chromium Single Cell Multiome ATAC + Gene Expression kit from 10x genomics. We targeted 5000 nuclei per mouse per group; single nuclei were isolated from snap-frozen isolated islet pellets from AL, CR, and HFD mice after 2 months on diet. Nuclei were isolated using the standard 10x Genomics protocol for nuclei isolation from embryonic mouse brain. Briefly, the islet pellet was lysed and homogenized in 0.1x Lysis Buffer solution (Tris-HCl 10 mM, NaCl 10 mM, MgCl$_2$ 3 mM, Tween-20 0.01%, IGEPAL CA-630 0.01%, Digitonin 0.01%, BSA 1%, DTT 1 mM, RNase inhibitor 1 U/ul). Then, the lysate was incubated on ice for 5 min, pipette mixed and incubated on ice for additional 10 min. Next, chilled wash buffer (Tris-HCl 10 mM, NaCl 10 mM, MgCl$_2$ 3 mM, Tween-20 0.1%, BSA 1%, DTT 1 mM, RNase inhibitor 1 U/ul) was added to the sample, pipette mixed, and the sample was filtered through a Flowmi Cell Strainer. Then, the pellet was washed with chilled wash buffer and the nuclei concentration was determined using Trypan Blue Stain (0.4%) and hemocytometer. The final nuclei concentration (1000 nuclei/ul) was resuspended in chilled Diluted Nuclei Buffer (10x Genomics) and immediately processed for transposition, barcoding, library construction and sequencing according to 10x Genomics Guidelines at Vanderbilt Technology for Advanced Genomics Core (VANTAGE).

## Analysis of single cell multiomics data

Raw sequencing reads were analyzed using Cell Ranger ARC Count v2.0.1 via 10x Genomics Cloud Analysis using default parameters and the mm10 2020-A-2.0.0 reference genome. Fragment and alignment files were input into AMULET (v1.1)[88] to detect multiplet droplets with mm10 repetitive elements from the UCSC genome browser. SoupX (v1.6.0)[89] was used to remove ambient RNA signal detected in empty droplets from nuclei-containing droplets. The contamination fraction of ambient RNA was estimated using the autoEstCont function. Seurat V4[90] objects containing assays for gene expression and chromatin accessibility were created for each sample and filtered for multiplets detected by AMULET. Nuclei with 1000-25000 UMIs for gene expression and 1000-100000 UMIs for ATAC peaks were retained. Nuclei were also filtered for nucleosome signal <2, transcriptional start site enrichment > 1, and percent mitochondrial RNA counts > 10% using Signac (v1.9.0)[91]. DoubletFinder (v2.0)[92] was used to detect heterotypic droplet doublets based on gene expression results. Peak calling was performed using the MACS (v2.2.7.1)[93] peak caller. Gene expression counts were normalized using the SCTransform[94] function in Seurat and PCA was calculated using the first 30 dimensions. For ATAC peaks,

the top features were identified using the FindTopFeatures function in Signac with min.cutoff = 5. ATAC counts were normalized using the RunTFIDF function and dimensionally reduced using the RunSVD function with default settings in Signac. TF motifs were identified based on the JASPAR2020[95] database using the AddMotifs function in Signac. Independent biological replicates were integrated by first merging Seurat V4 objects and then performing batch correction using Harmony (v0.1)[96] and uniform manifold approximation and projection (UMAP) was calculated using the first 20 dimensions from Harmony with the RunUMAP function from Seurat. Batch correction was also performed for the ATAC peaks using reciprocal latent semantic indexing (rLSI) in Signac[90,91]. For visualization, UMAP calculated from weighted nearest neighbors (wNN) was performed using the FindMultiModalNeighbors function in Seurat. Clustering was performed using the Louvain algorithm implemented in Seurat (resolution = 0.7), and cell types were annotated based on established pancreatic cell markers and UMAP visuals of marker genes. Beta cells (*n* = 16,555 cells) was further subset and reprocessed for cluster generation (Fig. 2J) performed using the Louvain algorithm with 10 PCs (resolution = 0.3). Differential expression between beta cell subclusters was performed using the FindAllMarkers function with only.pos = TRUE in Seurat.

## SCENIC and GRN reconstruction

To infer the GRN of islets per diet group, we leveraged pySCENIC[43]. This protocol allows the reconstruction of regulons (TFs and their known target genes) from single cell-derived co-expression data, assesses regulon activity at single cell resolution and allows the detection of regulon-enriched cell populations. Specifically, we ran v0.11.0 of pySCENIC as a Singularity container on ACCRE, Vanderbilt's High Performance Computing cluster. Following quality control and feature selection performed in gene expression assay from single cell multiomics data, the final object's raw counts and barcodes were converted to a LOOM file. Alongside of a list of 1721 mouse TFs (source: https://resources.aertslab.org)[43], this gene expression matrix served as input for calculating gene co-expression modules, via GRNBoost2 – a fast GRN inference algorithm that uses stochastic Gradient Boosting Machine regression with early-stopping regularization (Friedman 2002) and is made available through Dask. To control for the stochasticity associated with GRNBoost2, we calculated the co-expression modules 100 times and then retained only associations between TF and target genes that existed at least 80% of the time. We then merged the results of these 100 runs via a left outer join operation and averaged the Importance Metric values reported for each association. This consensus GRN was then used as input for module pruning, where we filtered out indirect gene targets lacking the cis-regulatory motif associated with each TF. This step relied on RcisTarget[42] and ranking databases for motifs (mm9-tss-centered-5kb-7species.mc9nr.feather) 10 kb around the TSS (+/−5kbp, mm9-tss-centered-5kb-7species.mc9nr.feather). The resulting co-expressed TF-target genes are then grouped into regulons. Finally, the per-cell activity of the regulons was computed using AUCell[42], which uses the "Area Under the Curve" (AUC) to calculate whether a subset of the input gene set is enriched within the expressed genes for each cell. This activity data was further binarized (assigned an "on" or "off" value, per regulon, per cell) by threshold on the AUC values of the given regulon. Both the AUCell and binarized regulon activity matrices were integrated into Seurat object via the CreateAssayObject, for downstream analysis and visualization.

## In vivo labeling of mice with $^{15}$N

$^{15}$N-labeled mice for MIMS-EM were generated as previously described[4]. Briefly, female mice were fed with $^{15}$N-enriched diet ( > 98% $^{15}$N enrichment, Cambridge Isotope Labs) for 70 days ad libitum (ad-lib) to saturate their system with $^{15}$N-labeled nutrients. Next, females were mated with a male for 1 week; pregnant females were maintained on $^{15}$N diet

during gestation to create [15]N-labeled pups. [15]N-diet was continuously provided during the lactation period until [15]N-pups were weaned at 21 days (P21) of age. Next, [15]N-pups were maintained on [15]N-diet feeding ad-lib until they age P45. Finally, [15]N-labeled mice were randomly allocated to AL, CR, or HFD feeding groups using normal chow rich in [14]N instead of [15]N (AL and CR groups received Rodent Diet 5053, 4.11 Kcal/g; HFD mice were fed with Rodent Diet 58Y1, 5.10 Kcal/g) for up to 12 months. In addition, one P60 [15]N-mouse was euthanized before introduction of the [14]N-rich feeding to determine the maximum levels of 15 N enrichment. This information is needed to calculate the number of cell division cycles based on the dilution of the [15]N retained inside each cell nucleus[97,98].

## Sample processing for MIMS-EM

[15]N-tissue processing and MIMS-EM imaging were performed as previously established by us[4,7,8]. After 12 months on diet, [15]N-mice in AL, CR, or HFD diet groups were euthanized and perfused for 30 seconds with Ringer's solution (0.79% NaCl/0.038% KCl/0.02% $MgCl_2 \cdot 6H_2O$/0.018% $Na_2HPO_4$/0.125% $NaHCO_3$/0.03% $CaCl_2 \cdot 2H_2O$/0.2% dextrose/0.02% xylocaine) followed by a transcardiac perfusion with fresh 2.5% glutaraldehyde and 2% PFA in 0.15 M sodium cacodylate for 10 min (rate at 5 mL/min). Next, the pancreas was dissected and prepared for X-ray microscopy (XRM) and scanning electron microscopy (SEM) followed by MIMS. Perfusion-fixed [15]N-tissue was cut into ~1mm[3] pieces and then postfixed in the same fixative at 4 °C overnight. Next, each sample was washed using 0.15 M cacodylate buffer and then fixed in 2% osmium tetroxide and 1.5% potassium ferrocyanide in 0.15 M sodium cacodylate buffer for 1 hour at room temp. After that, samples were washed extensively using double distilled water ($ddH_2O$) and placed in a 0.5% thiocarbohydrazide solution for 30 min followed by five washes in $ddH_2O$. Next, tissue samples were placed in a 2% aqueous osmium tetroxide solution for 1 hour, thoroughly washed in $ddH_2O$, and placed in a 2% aqueous uranyl acetate solution at 4 °C for overnight incubation. Samples were then washed with $ddH_2O$ and placed into Walton's lead aspartate solution for 30 min and baked at 60 °C in a bench-top oven. Following the bake, samples were washed with $ddH_2O$ followed by serial dehydration using ice-cold 70% EtOH, 90% EtOH, 100% EtOH, 100% EtOH, and dry acetone (10 min each step on ice). Next, each tissue piece was placed into 1:3, 1:1, and 3:1 solutions of Durcupan ACM:acetone for 12 h in each concentration. Finally, the tissue was exposed to three changes of 100% Durcupan ACM for 24 h each before being baked for 48 h at 65 °C for solidification.

## Identification and mapping of in situ islets using X-ray microscopy

Identification of the location of endogenous islets in our EM-ready tissue blocks was achieved using X-ray Microscopy (XRM) as previously established by us[4]. Here, epoxy-embedded pancreas blocks were scanned using XRM (X-radia Versa 510, Zeiss) with the source set at 80 kV and 7 W power using a 4X power objective (final pixel size ~ 2 microns depending on final geometric magnification). The specimens were rotated 360 degrees and 3D projection images were collected per tissue block. Islets were identified based on their x-ray contrast and their 3D coordinates acquired using XRM to guide tissue and islet sectioning using a Leica ultramicrotome to trim the block and cut 80-nm-thick sections for imaging[4]. Sections were placed on 10x10mm or 5x7mm silicon wafers (Electron Microscopy Sciences (EMS), USA) for downstream SEM, tomography, and/or MIMS imaging.

## Correlative electron microscopy and multi-isotope mass spectroscopy (MIMS-EM)

MIMS-EM measures spatially localized concentrations of several isotopes in biological samples overlaid with high-resolution scanning electron microscopy (SEM) to provide accurate spatial and quantitative information regarding the chemical composition of macromolecules, organelles, cells, and tissues[4,7,8]. Briefly, MIMS-EM images of brain neurons and islets were acquired using 80-nm-thick sections arranged on silicon wafers. Cells were mapped using large field of view SEM using a GeminiSEM or MerlinSEM (Zeiss, Germany) guided by automated tile acquisition and image mosaicking using Atlas 5 software (Fibics, Ottawa, Canada). Images were acquired using a pixel size of 5 nm, and we covered areas of approximately 300um[2] per tissue. Next, the same samples were transferred to a MIMS microscope (50 L SIMS, Cameca, France) and acquisition of isotope maps ([15]N, [14]N, [12]C, and [32]S) was achieved using a cesium (Cs − ) beam. MIMS image frames contained 512 × 512 pixels to cover a raster of 30-to-40um[2], and we acquired at least three frames per raster using 10 min imaging per frame and used the beam adaptor D1-3.

## MIMS-EM image registration and analysis

MIMS and EM images were co-registered and overlaid as recently described[8]. Briefly, the [32]S chemical maps obtained by MIMS were used to create spatial fiducials that were projected on the SEM image to create image alignment matrices. These matrices were then applied to ratiometric [15]N-to-[14]N images, which relay cell age information. Prior to alignment and registration, our MIMS images were processed using default configurations using the OpenMIMS plug in in ImageJ[99]. Quantification of mouse beta cell and neuron age was also done using OpenMIMS, where we determined the [15]N/[14]N ratios in each cell nucleus. Proliferating cells lose 50% of nuclear [15]N after each cell division[97] and therefore we used this principle to determine the relative age and number of division cycles for each beta cell in AL, CR, and HFD islets.

## Electron tomography (eTomo)

Semi-thick sections of thickness about 300 nm were cut from the blocks of cells prepared for SEM with a Leica ultramicrotome and placed on 100-mesh copper grids. 20-nm colloidal gold particles were deposited on each side of the grid to serve as fiducial cues. The grids were placed in a single-rotation tilt holder and inserted in a Tecnai High Base Titan (FEI; Hillsboro, OR) electron microscope operated at 300 kV. The grids were irradiated with electrons for about 10 min to limit anisotropic specimen thinning during image collection at the magnification used to collect the tilt series before initiating a dual-axis tilt series. During data collection, the illumination was held to near parallel beam conditions and the beam intensity was kept constant. Tilt series were captured using SerialEM software (University of Colorado, Boulder, CO). Tilt series were taken at 0.81 nm/pixel. Images were recorded with a Gatan Ultrascan 4Kx4K CCD camera. Each dual-axis tilt series consisted of first collecting 121 images taken at 1 degree increment over a range of −60 to +60 degrees followed by rotating the grid 90 degrees and collecting another 121 images with the same tilt increment. After collecting the orthogonal tilt series, to improve the signal-to-noise ratio, 2x binning was performed on each image by averaging a 2 × 2 x-y pixel box into 1 pixel using the "newstack" command in IMOD (University of Colorado, Boulder, CO). The IMOD package with etomo java wrapper (https://en.wikipedia.org/wiki/IMOD) was used for tilt-series alignment, reconstruction, and volume segmentation. R-weighted back projection was used to generate the reconstructions. The two volumes were combined using the common fiducial markers shared by each volume and patch-matching using a warping residual below 0.35 aided the combination.

The biophysical modeling schema of[58] was used to estimate the potential rate of ATP production by mitochondria using a model for condensed mitochondria; the respiratory control ratio (RCR) is needed to convert from condensed to the orthodox mitochondria of beta cells. The RCR is defined as state 3 mitochondrial respiration (condensed) divided by state 4 mitochondrial respiration (orthodox). The RCR was

provided by the Long et al 2019. Mouse heart interfibrillar mitochondria had an RCR of about 4.7. For purposes of this modeling, we assume that beta cell mitochondria RCR is close to heart RCR. This modeling provided similar rates to those published using different models[100–102].

## Islet cell type classification in SEM maps

Beta cells were classified according to the classical morphology of insulin granules in EM micrographs with their condensed electron-dense nature of insulin crystals surrounded by a large halo[103].

## Imaging mass spectrometry

Fresh frozen mouse pancreas tissue was sectioned at 12 µm thick on a cryostat (Leica Biosystems). MALDI matrix (9-aminoacridine, 9AA) (Sigma-Aldrich) was spray-coated onto the target slides in an automated fashion using a TM Sprayer (HTX Imaging). 9-AA was made up as a 5 mg/ml solution in 90% methanol. Four passes were used with a nozzle temperature of 85°, a flowrate of 0.15 ml/min, 2-mm track spacing, and a stage velocity of 700 mm/min. Nitrogen was used as the nebulization gas and was set to 10 psi. Images were acquired on a 15 T Fourier transform ion cyclotron resonance mass spectrometer (FT-ICR MS, Solarix, Bruker Daltonics) equipped with an Apollo II dual ion source and Smartbeam II 2 kHz Nd:YAG laser that was frequency tripled to 355 nm. Data were collected in the negative ion mode with the laser operating at 2 kHz at 50 µm spatial resolution. Tentative metabolite identifications were made by accurate mass, typically better than 1 ppm. Ion mass resolution was of $\pm 0.00565$ Da. Images were analyzed with flexImaging and SCiLS Lab software (Bruker) using the ROC method with automatic peak detection to identify a total of $n = 290$ metabolites from the combined mass spectra. An area under the curve threshold of $0.5 \pm 0.1$ was used to define metabolites that were enriched in CR ($n = 52$ metabolites) or in AL ($n = 28$) islet regions, respectively (Fig. 4C, Supp Tables 8, 9).

## In situ hybridization

RNAscope multiplex fluorescent assay was performed on FFPE pancreas slices, according to the manufacturer's instructions (Advanced cell Diagnostics, Cat. No. 320293). Briefly, slides were heated at 60 °C for 20 min and then deparaffinized in xylene and 100% ethanol. After drying, the slides were incubated with $H_2O_2$ for 10 min and then washed in DiH$_2$0. Next, slides were boiled in antigen retrieval solution (>98 °C) for 15 min, washed in DiH$_2$0, dehydrated in 100% ethanol, and incubated with Protease Plus for 30 min at 40 °C. After washing, slides were incubated with probes mixed for 2 h at 40 °C. RNAscope® *Cdkn2a/p16* and *Cdkn1a/p21* probes (Advanced Cell Diagnostics, #411011-C1 and #408551-C2, respectively) were used. All slides were incubated overnight in 5× SSC buffer (Invitrogen, #15557044) and then treated with a series of specific signal amplifications for each probe. Probes were finally detected using the Opal dyes 570 and 650, individually diluted in TSA buffer and incubated for 30 min at 40 °C each. Next, slides were washed in PSB, incubated with blocking solution (3% BSA, 0.3% Triton X-100 in PBS) for 30 min and then kept overnight at room temperature with primary antibody (guinea pig anti-insulin at 1:4 dilution). Then, slides were washed and incubated with DAPI and secondary antibody (Alexa Fluor 488,1:400) at room temperature for 1 h. Finally, stained slides were washed again and mounted in Vecta-Shield prior to confocal imaging. RNAscope image analysis was performed by applying a machine-learning segmentation model trained using RNAscope data from multiple RNAscope targets. Here, analyzed images are intensity normalized, and images containing algorithm-detected RNAscope spots were thresholded and cleaned using median filters. Next, the expression of RNAscope targets *p16* and *p21* was analyzed using CellProfiler to quantify the number of *p16* or *p21* spots per beta cell area. Beta cells were identified by the pixel intensity of the Insulin channel.

## Statistics

Statistical analysis of RNA expression and ATAC peak levels was performed within Seurat using default parameters. For animal experiments and IHC microscopy data, Student's *t* test (Prism 9, GraphPad) was used to compare two groups, whereas a one-wat ANOVA was used to compare two or more groups. A *p* value of <0.05 was considered statistically significant.

## Reporting summary

Further information on research design is available in the Nature Portfolio Reporting Summary linked to this article.

## Data availability

Tabular and processed sequencing data is deposited in Zenodo (https://doi.org/10.5281/zenodo.6491943). The single nucleus multiome sequencing data has been deposited in the GEO repository (GSE276572, https://www.ncbi.xyz/geo/query/acc.cgi?acc=GSE276572). Source data are provided with this paper.

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

## Acknowledgements

We are thankful to Michelle Reyzer from the Vanderbilt Mass Spectrometry Research Center for assistance with MALDI-MS imaging and analysis, to the outstanding team at the Vanderbilt Mouse Metabolic Phenotyping Center for all the assistance with in vivo glucose homeostasis tests (DK135073, 1S10RRO28101-01). We also thank the University of Michigan Animal Phenotyping Core for conducting the bomb

calorimetry experiments (1U2CDK110768, DK020575, and DK089503). This research was supported by recruitment funds from the Vanderbilt's Department of Molecular Physiology and Biophysics and NIH grants 1R03DK127484, 5U24DK097771, and 1R01DK138141 to RAeD, by a grant from the Canadian Institutes of Health Research (CIHR; 487188) to PEM, by and NIH DK132669 to D.D. PEM holds the Canada Research Chair in Islet Biology.

## Author contributions

C.D.S., A.C., M.H., M.H.E., and R.Ae.D. designed experiments; C.D.S., A.C., S.S., M.C., G.P., V.L.R., C.A., B.R., K.K., T.D., M.C., D.D., J.P.C., P.E.M., M.H., M.H.E., and R.Ae.D. collected and analyzed data. C.D.S. and R.Ae.D. wrote the manuscript with significant input from all authors.

## Competing interests

The authors declare no competing interests.
