## [Peer Review File · Nature Communications]

Calorie restriction increases insulin sensitivity to promote beta cell homeostasis and longevityREVIEWER COMMENTS

Reviewer #1 (Remarks to the Author):

In this manuscript, Santos et. al well combined single-cell multiomics and multi-modal high-resolution microscopy approaches to investigate how CR modulates beta cell heterogeneity and longevity. The analyses utilized for this study clearly demonstrate the transcriptional reorganization and mitochondrial function enhancement of beta cells in response to CR. However, there are some concerns that must be addressed for this paper to be publishable.

#1. In the Results section, the descriptions for Figure 1F and Figure 1H-J appear somewhat similar. I recommend that the authors consider utilizing the calculated values presented in either Figure 1H or Figure 1I to enhance the description of their findings, which will provide readers with a more precise understanding of the influence of CR on increased insulin sensitivity.

#2. In the section discussing imaging mass spectrometry, the authors mention that the reason for utilizing MSI is to minimize changes in cell metabolism that may have occurred during the islet isolation process. While this is a valid point, it's important to emphasize that MSI has the unique capability to elucidate and visualize metabolic heterogeneity within tissues, which offers another significant advantage that should be highlighted in the text.

#3. In the part where the authors compare ion differences between islets and acinar compartments, as well as between AL and CR islets, I wonder why they opted for the receiver operating characteristic (ROC) method. It's more common to employ univariate methods like OPLS-DA or tSNE for such comparisons. I also suggest providing additional details about this methodology in the SI section.

#4. The author had identified a variety of metabolites using the accurate mass, however, whether the current dataset support precise location information for identified metabolites, particularly signaling molecules like uridine 2',3'-cyclic phosphate or fructose-6-phosphate? If conditions permit, the use of adjacent tissue slices for secondary mass spectrometry could enhance the conclusiveness of the findings regarding changes in insulin release metabolism.

#5. Regarding Figure 4E, the classification of enriched metabolites had raised some confusion as certain signaling molecules might also be lipids. I suggest adopting a more straightforward and non-overlapping categorization, such as amino acids, lipids, carbohydrates, and nucleotides.

#6. In the part comparing islets versus acinar compartments, the authors mention several ions of interest; however, only one ion is presented in Fig 4G (m/z: 465.3044). To enhance the comprehensibility of the findings, I recommend that the authors consider supplementing the pictures in the main manuscript or the SI with corresponding mass spectrometry imaging data for the other ions mentioned.

#7. The authors have devised an exceptional experiment to assess beta cell longevity. However, given the previously established gender differences in CR-induced insulin sensitivity, I'm curious whether the authors have taken into account this gender-related influence in their study involving P45 mice?

#8. There are notable areas where formatting and language expressions could be improved. For example, uniform decimal places for m/z values in the imaging mass spectrometry part.

Reviewer #2 (Remarks to the Author):

This is an interesting and overall well executed study to document the effects of caloric restriction on beta cell health, identity and longevity. The authors demonstrate that beta cell rest or reduced metabolic demand achieved by reduced caloric input preserves peripheral insulin sensitivity, which then reduces demand on the beta cell. This – perhaps predictably – reduces beta cell turnover accounting for an increased proportion of long-lived cells. Many of the transcriptional changes follow those observed by others. The strength of this paper lies in the comprehensive series of approaches to document in considerable detail how beta cells respond favorably to the reduced metabolic demands of caloric restrictions. The results in part validate prior work, and extend considerably by the good use of isotope labeling. There is considerable strength in the combination of comprehensive series of approaches, executed carefully and skillfully.

Specific comments:

Figure resolution was quite poor. This makes it hard to assess the quality of the staining for Lamp1 etc. Labels for the network analysis were not legible. It looks like the files as uploaded were of low resolution. Supplementary files were of much better quality

beta cell mass was not determined to be unchanged (Figure 1), which appears at slight odds with the observation of increased number of long-lived cells, which indicates reduced beta cell turnover and expansion in the early adult phase (p45 – adult). Perhaps the authors can help clarify here?

The sex-dependent nature of the phenotype described at the top of page 4 was not entirely clear.

-the observations of reduced cAMP in CR in Figure 1 appear to conflict with the increased cAMP metabolite detected in CR in Figure 4.

-the increased detection of ADP and cyclic ADP ribose are not sufficient to conclude DNA damage. Direct detection of single strand breaks, or increased DNA repair markers would be necessary to support such a conclusion. The alternative could be to simply strike the statement.

Reviewer #3 (Remarks to the Author):

In this manuscript, dos Santos et al. describe how caloric restriction (CR) modulates pancreatic beta cell function through transcriptional regulation of gene regulatory networks (GRNs). These transcriptional changes affect autophagy, mitophagy and mitochondrial function, DNA damage, and senescence, which is associated with increased efficiency and longevity of beta cells. They also show that long term CR (12 months) reduces beta cell turnover. The experimental design is simple and straightforward, though it would be helpful for the readers if authors discuss further the inclusion of high-fat diet (HFD) fed mice in

some of the experiments. The study successfully describes the changes triggered by CR in beta cells. However, data provided do not fully address whether there is a direct cause-effect relationship between CR and the changes observed in beta cells. As authors discuss, it seems that increased insulin sensitivity is a key mediator that leads to transcriptional remodeling, and that the increase in beta cell function and longevity seems to be a consequence of improved peripheral insulin sensitivity, rather than CR itself. Because of this, there are several statements throughout the manuscript that may need to be tuned down. For instance, the title (Caloric restriction promotes beta cell longevity and delays aging and senescence by enhancing cell identity and homeostasis mechanisms) should be rephrased to avoid the assumption of causality and summarize their results more accurately. Is this reviewer's opinion that the results presented in this manuscript are not convincing enough to conclude that CR directly enhances beta cell identity, nor that enhanced beta cell identity promotes beta cell longevity. Results rather suggest that this remodeling is a consequence of improved glucose homeostasis due to decreased insulin resistance, which lowers the demand of insulin production. Some ideas to address these points could be:

- to investigate if enhancing insulin sensitivity without CR gives the same outcome.
- to study whether CR remodels beta cell GRNs and induce the expression of beta cell identity genes in irreversibly insulin resistant mice.
- describe if starvation of beta cells in culture triggers the overexpression of beta cell identity genes.

The introduction section could benefit from some rephrasing when introducing the role of beta cell aging in the development of type 2 diabetes. For instance, authors declare 'Loss of beta cell insulin secretion can occur during normal aging and is the cause of type 2 diabetes'. This statement can be misleading; though it is true that aging can affect beta cell function, increased type 2 diabetes in older adults seems to be related to unhealthy lifestyle rather than aging per se. As stated in Basu et al., 'We conclude that the deterioration in glucose tolerance that occurs in healthy elderly subjects is due to a decrease in both insulin secretion and action with the severity of the defect in insulin action being explained by the degree of fatness rather than age per se.', and 'The effects of age on β -cell function has been a matter of debate with previous investigators reporting an increase (2,15,16), decrease (1,7,8,11,14), or no change (3,4,10,12,13) in insulin secretion in the elderly.'

The importance of beta cell longevity for beta cell function is an interesting discovery that could be explored further. Some questions that authors could address are:

- are CR mice more resistant to insulin resistance-induced beta cell dysfunction?
- what is the role of LLCs in HFD mice?
- do LLCs have increased insulin production compared to 'young' beta cells (product of proliferation)?
- are LLCs resistant to exhaustion, apoptosis, or senescence during aging?

Minor points

- Main figures are blurry and pixelated, and some schemes and images with greater detail were very difficult to evaluate.

- Definition of the scale in histology pictures is missing in several figure legends.
- The number of mice used for measurements in each panel is missing in several figure legends.
- Immunofluorescence of Lamp1 and Sdha in Figure 6 looks completely negative. Same comment for Lamp1 in Figure S8. Maybe a technical issue with color definition when exporting the images or converting them to PDF?

1. Rita Basu, Elena Breda, Ann L. Oberg, Claudia C. Powell, Chiara Dalla Man, Ananda Basu, Janet L. Vittone, George G. Klee, Puneet Arora, Michael D. Jensen, Gianna Toffolo, Claudio Cobelli, Robert A. Rizza; Mechanisms of the Age-Associated Deterioration in Glucose Tolerance: Contribution of Alterations in Insulin Secretion, Action, and Clearance. *Diabetes* 1 July 2003; 52 (7): 1738–1748. <https://doi.org/10.2337/diabetes.52.7.1738>

REVIEWER COMMENTS

We thank the reviewers for their insightful comments and constructive feedback. We have performed additional experiments and revised our manuscript to address the reviewers' concerns.

Reviewer #1 (Remarks to the Author):

In this manuscript, Santos et. al well combined single-cell multiomics and multi-modal high-resolution microscopy approaches to investigate how CR modulates beta cell heterogeneity and longevity. The analyses utilized for this study clearly demonstrate the transcriptional reorganization and mitochondrial function enhancement of beta cells in response to CR. However, there are some concerns that must be addressed for this paper to be publishable.

Thank you, we have addressed these concerns throughout the text and our answers are outlined below.

#1. In the Results section, the descriptions for Figure 1F and Figure 1H-J appear somewhat similar. I recommend that the authors consider utilizing the calculated values presented in either Figure 1H or Figure 1I to enhance the description of their findings, which will provide readers with a more precise understanding of the influence of CR on increased insulin sensitivity.

We have re-written that section to make it more concise and highlight our results regarding glucose homeostasis mechanisms of CR mice.

#2. In the section discussing imaging mass spectrometry, the authors mention that the reason for utilizing MSI is to minimize changes in cell metabolism that may have occurred during the islet isolation process. While this is a valid point, it's important to emphasize that MSI has the unique capability to elucidate and visualize metabolic heterogeneity within tissues, which offers another significant advantage that should be highlighted in the text.

Thank you, we agree with your comment. The text has been revised to highlight this point.

#3. In the part where the authors compare ion differences between islets and acinar compartments, as well as between AL and CR islets, I wonder why they opted for the receiver operating characteristic (ROC) method. It's more common to employ univariate methods like OPLS-DA or tSNE for such comparisons. I also suggest providing additional details about this methodology in the SI section.

We chose the ROC method because an ROC curve allows for quantification in the variability of individual/detected ions between AL or CR conditions (islets or acinar compartments). The ROC method tends to deliver more robust and easier results to interpret than multivariate or visual clustering methods patterns like t-SNE/UMAPs (which would require optimization of different parameters). In this revised manuscript version, we have incorporated additional ion panels in the main figure 5 and supplementary figure 7. In addition, Figure S7C now includes a 3D component plot after our MALDI-MS data was analyzed using pSLA.

#4. The author had identified a variety of metabolites using the accurate mass, however, whether the current dataset support precise location information for identified metabolites, particularly signaling molecules like uridine 2',3'-cyclic phosphate or fructose-6-phosphate? If

conditions permit, the use of adjacent tissue slices for secondary mass spectrometry could enhance the conclusiveness of the findings regarding changes in insulin release metabolism.

We have considered the possibility of running secondary mass spec of laser-dissected islets to confirm the identity and enrichment of the metabolites identified with MALDI-MS, however this proved extremely challenging due to the low islet yield acquired.

#5. Regarding Figure 4E, the classification of enriched metabolites had raised some confusion as certain signaling molecules might also be lipids. I suggest adopting a more straightforward and non-overlapping categorization, such as amino acids, lipids, carbohydrates, and nucleotides.

Thank you for this suggestion; the potential overlap between molecular classes is now addressed in the manuscript text. Due to space constraints, this panel has been moved to supplementary figure 7.

#6. In the part comparing islets versus acinar compartments, the authors mention several ions of interest; however, only one ion is presented in Fig 4G (m/z: 465.3044). To enhance the comprehensibility of the findings, I recommend that the authors consider supplementing the pictures in the main manuscript or the SI with corresponding mass spectrometry imaging data for the other ions mentioned.

The figure has been revised and we now include additional panels highlighting multiple ions.

#7. The authors have devised an exceptional experiment to assess beta cell longevity. However, given the previously established gender differences in CR-induced insulin sensitivity, I'm curious whether the authors have taken into account this gender-related influence in their study involving P45 mice?

Thank you. Unfortunately, ¹⁵N-labelling and MIMS-EM are very expensive techniques and therefore we have not been able to evaluate beta cell longevity in females.

#8. There are notable areas where formatting and language expressions could be improved. For example, uniform decimal places for m/z values in the imaging mass spectrometry part.

This has been addressed and the manuscript text has been revised.

Reviewer #2 (Remarks to the Author):

This is an interesting and overall well executed study to document the effects of caloric restriction on beta cell health, identity and longevity. The authors demonstrate that beta cell rest or reduced metabolic demand achieved by reduced caloric input preserves peripheral insulin sensitivity, which then reduces demand on the beta cell. This – perhaps predictably – reduces beta cell turnover accounting for an increased proportion of long-lived cells. Many of the transcriptional changes follow those observed by others. The strength of this paper lies in the comprehensive series of approaches to document in considerable detail how beta cells respond favorably to the reduced metabolic demands of caloric restrictions. The results in part validate prior work and extend considerably by the good use of isotope labeling. There is considerable strength in the combination of comprehensive series of approaches, executed carefully and skillfully.

Specific comments:

1) Figure resolution was quite poor. This makes it hard to assess the quality of the staining for Lamp1 etc. Labels for the network analysis were not legible. It looks like the files as uploaded were of low resolution. Supplementary files were of much better quality.

Thank you for pointing this out. We believe the upload process significantly reduced the figure quality. We have re-worked our figure panels and figures are uploaded with high-resolution.

2) Beta cell mass was not determined to be unchanged (Figure 1), which appears at slight odds with the observation of increased number of long-lived cells, which indicates reduced beta cell turnover and expansion in the early adult phase (p45 – adult). Perhaps the authors can help clarify here?

The reviewer raises a good point. To better investigate this point, we performed a side-by-side comparison of beta cell mass measurements from 2 different and independent 2- and 12-mo old cohorts – the compiled data is shown below. We found a significant ~2-fold increase in the overall beta cell mass over time in AL mice, which supports the expected continuous growth of beta cell mass over time, whereas CR mice have only a ~1.4-fold increase trend. Importantly, the difference between the beta cell mass of 2-month vs 12-month-old cohorts was near-significant in terms of raw beta cell mass (graph on the left) - and reached significance when the data was normalized to the respective 2mo cohort (graph on the right). The graph on the left with the raw beta cell mass measurements has been incorporated into a revised version of Figure S1G.

3) The sex-dependent nature of the phenotype described at the top of page 4 was not entirely clear.

We have revised this section of the manuscript to add context from recent literature.

4) The observations of reduced cAMP in CR in Figure 1 appear to conflict with the increased cAMP metabolite detected in CR in Figure 4.

We believe the reviewer is referring to the contrast between Liraglutide, islet perfusion, and MALDI-MS experiments. Based on the in vitro experiments, we expected CR beta cells to have reduced cAMP, however this result is contradicted by the liraglutide MTT and MALDI-MS results. We speculate that the in vitro nutrient-rich conditions required for islet culture and experimentation are likely modulating CR beta cell mechanisms to produce relatively artificial and in vitro only effects.

-the increased detection of ADP and cyclic ADP ribose are not sufficient to conclude DNA damage. Direct detection of single strand breaks, or increased DNA repair markers would be necessary to support such a conclusion. The alternative could be to simply strike the statement.

*We have revised this section and toned down our conclusions: "We hypothesized CR would delay and/or prevent the onset of beta cell aging signatures such as accumulation of DNA damage **markers** and expression of senescence-associated genes. To test this hypothesis, we maintained 8-week-old male mice to CR for 12 months; this led to an overall lower body weight with reduced fat content, arrested pancreas growth, lower beta cell mass, and overall lower meal-stimulated beta cell insulin release due to increased peripheral insulin sensitivity (Figure 6A-H, Figures S1G and S7A). To investigate the effect of CR on beta cell aging hallmarks in situ, we performed immunohistochemistry (IHC) and confocal microscopy of mouse pancreases from AL or CR mice after 2- or 12-months. First, **we observed reduced accumulation of the DNA damage marker 53bp1 in CR beta cells, which suggests that these cells might accumulated less age-related DNA damage (Figure 6I).**"*

Reviewer #3 (Remarks to the Author):

In this manuscript, dos Santos et al. describe how caloric restriction (CR) modulates pancreatic beta cell function through transcriptional regulation of gene regulatory networks (GRNs). These transcriptional changes affect autophagy, mitophagy and mitochondrial function, DNA damage, and senescence, which is associated with increased efficiency and longevity of beta cells. They also show that long term CR (12 months) reduces beta cell turnover. The experimental design is simple and straightforward, though it would be helpful for the readers if authors discuss further the inclusion of high-fat diet (HFD) fed mice in some of the experiments. The study successfully describes the changes triggered by CR in beta cells. However, data provided do not fully address whether there is a direct cause-effect relationship between CR and the changes observed in beta cells. As authors discuss, it seems that increased insulin sensitivity is a key mediator that leads to transcriptional remodeling, and that the increase in beta cell function and longevity seems to be a consequence of improved peripheral insulin sensitivity, rather than CR itself. Because of this, there are several statements throughout the manuscript that may need to be tuned down. For instance, the title (Caloric restriction promotes beta cell longevity and delays aging and senescence by enhancing cell identity and homeostasis mechanisms) should be rephrased to avoid the assumption of causality and summarize their results more accurately.

Is this reviewer's opinion that the results presented in this manuscript are not convincing enough to conclude that CR directly enhances beta cell identity, nor that enhanced beta cell identity promotes beta cell longevity. Results rather suggest that this remodeling is a consequence of improved glucose homeostasis due to decreased insulin resistance, which lowers the demand of insulin production. Some ideas to address these points could be:

- to investigate if enhancing insulin sensitivity without CR gives the same outcome.

- to study whether CR remodels beta cell GRNs and induce the expression of beta cell identity genes in irreversibly insulin resistant mice.
- describe if starvation of beta cells in culture triggers the overexpression of beta cell identity genes.

Thank you for your comments and suggestions. We agree with the reviewer that testing for the importance of enhanced peripheral insulin sensitivity in our CR model is an important point worth investigating further. Towards this goal, we conducted new experiments that 1) establish/confirm the temporal co-incidence of the CR-associated beta cell secretory phenotype with increased insulin sensitivity (Figure S2I-K), and 2) using a previously published model of 20% calorie dilution (DL) that minimizes fasting ¹, we show that DL is insufficient to enhance insulin sensitivity, or modulate beta cell function and mTOR/autophagy pathways (vs CR, Figure 1 and S7). Therefore, we conclude that the “conventional” CR feeding regiment presented here (i.e., calorie restriction with time-restricted feeding and long periods of fasting that enhance insulin sensitivity) is required for the beneficial effects on beta cell structure-function. Finally, we have revised our manuscript title to summarize these findings: “Calorie restriction increases insulin sensitivity to promote beta cell homeostasis and longevity”.

Regarding your experimental suggestions (i.e., enhancing insulin sensitivity without CR, the effects of CR in insulin resistant mice, or an in vitro model of CR): Given that our new experiments reinforce the inter-dependency of insulin sensitivity and beta cell health (described above), we strongly believe that CR of genetically modified / insulin-resistant animals, or optimization of in vitro models of CR, are quests that are outside the scope of this manuscript. In addition, these experiments would require a significant amount of time to establish entirely new mouse colonies (e.g., Glut2KO or IRKO mice) or in vitro models (e.g., EndoCBH cells) in our lab/Vanderbilt facilities. These concepts and ideas will be explored in a near-future project in our lab, including mouse models of accelerated beta cell aging and impaired glucose homeostasis (i.e., MAFA S64F).

The introduction section could benefit from some rephrasing when introducing the role of beta cell aging in the development of type 2 diabetes. For instance, authors declare ‘Loss of beta cell insulin secretion can occur during normal aging and is the cause of type 2 diabetes’. This statement can be misleading; though it is true that aging can affect beta cell function, increased type 2 diabetes in older adults seems to be related to unhealthy lifestyle rather than aging per se. As stated in Basu et al., ‘We conclude that the deterioration in glucose tolerance that occurs in healthy elderly subjects is due to a decrease in both insulin secretion and action with the severity of the defect in insulin action being explained by the degree of fatness rather than age per se.’, and ‘The effects of age on β -cell function has been a matter of debate with previous investigators reporting an increase (2,15,16), decrease (1,7,8,11,14), or no change (3,4,10,12,13) in insulin secretion in the elderly.’¹

Thank you for pointing this out. We have re-phrased this paragraph to summarize the beta cell aging studies and conclude with: “Together, when combined with an unhealthy lifestyle during old age, these signatures could pre-dispose beta cells to failure and lead to age-associated onset of type 2 diabetes (T2D)”

The importance of beta cell longevity for beta cell function is an interesting discovery that could be explored further. Some questions that authors could address are:

- are CR mice more resistant to insulin resistance-induced beta cell dysfunction?

Great question. To address it, we exposed lean CR mice (which were kept on CR diet for 2 months) to HFD feeding for another 2 months (the CR-HF group). As expected HFD-fed mice become obese, and glucose intolerant / insulin resistant. We also show that CR does not grant any kind of metabolic protection/advantage. In fact, we uncovered that CR-HF beta cells have signs of impaired meal-induced insulin secretion, which is associated with down-regulation of several genes linked to cell proliferation and pancreas development/function (vs control-HF mice). Indeed, CR-HF beta cells have a compromised adaptive proliferative response during HFD, as revealed by the significantly lower beta cell mass of CR-HF pancreases. These results are described in a new section of the manuscript and shown in Figure 2.

- what is the role of LLCs in HFD mice?

We believe that LLCs could serve as a “beta cell reserve pool” that participates in the beta cell adaptive response. This concept has been added to the discussion section:

“Given that the share of long-lived beta cells decreases during HFD, we speculate that these cells could act as a “beta cell reserve pool” that participates in the adaptive beta cell proliferation response.”

- do LLCs have increased insulin production compared to ‘young’ beta cells (product of proliferation)?

Likely no, since CR beta cells (which are mostly LLCs) have similar insulin granule contents than AL beta cells (where ~60% are LLCs) (this data is shown in Figure S8C).

- are LLCs resistant to exhaustion, apoptosis, or senescence during aging?

We don’t know yet. To address these questions, we are currently developing correlative microscopy probes that are compatible with MIMS-EM so that we can measure beta cell age (to identify LLCs) and markers of apoptosis or senescence (as shown in this manuscript). These efforts are part of a separate and future projects that will showcase new modalities of MIMS-EM to the beta cell field.

Minor points

- Main figures are blurry and pixelated, and some schemes and images with greater detail were very difficult to evaluate.

- Definition of the scale in histology pictures is missing in several figure legends.

- The number of mice used for measurements in each panel is missing in several figure legends.

- Immunofluorescence of Lamp1 and Sdha in Figure 6 looks completely negative. Same comment for Lamp1 in Figure S8. Maybe a technical issue with color definition when exporting the images or converting them to PDF?

We apologize for these missteps. The low figure resolution was due to the initial submission process. We have re-generated the panels (to include the missing scale bars) and the PDFs have been saved with higher-resolution standards.

References cited:

1. Pak, H.H., Haws, S.A., Green, C.L., Koller, M., Lavarias, M.T., Richardson, N.E., Yang, S.E., Dumas, S.N., Sonsalla, M., Bray, L., et al. (2021). Fasting drives the metabolic, molecular and geroprotective effects of a calorie-restricted diet in mice. *Nature Metabolism* 3, 1327-+. [10.1038/s42255-021-00466-9](https://doi.org/10.1038/s42255-021-00466-9).

REVIEWER COMMENTS

Reviewer #1 (Remarks to the Author):

The authors addressed the majority of issues me and other reviewers have raised. The quality of the images has been greatly improved. Some minor clarifications should be warranted as pointed out below.

#1. In the section of “MALDI MS metabolomics”, the term “enriched” can be ambiguous. Does this imply a significant upregulation?

#2. Line 487, normal levels of metabolites are not suggested to show metabolic alternations.

#3. Could the author provide more details on the process of ROC curve analysis and the threshold setting?

#4. In Figure 5E, the AL enriched part, could the author specify the compound or type of the metabolite ions m/z 327.23271 and 766.53786? Based on the accurate mass, m/z 327.23271 is more likely a fatty acid rather than phosphatidylethanolamine.

#5. In Figure 5E, please enhance the descriptions in the legend to clearly indicate the locations of islets and acinar compartments. In addition, some professional terminology need to be improved. The expression of “766.53786 $m/z \pm 0.00565$ Da” is inappropriate, as m/z is a ratio without units and Da is a unit of atomic or molecular mass. Five ions are identified with same $\Delta m \pm 0.00565$ Da. Please verify this. If this represents the m/z tolerance, it can only retain one significant digit. Usually it is allowed not to be indicated in the figures.

#6. The author should provide information about the mass spectrometry resolution in MALDI MS experiments; typically, retaining four decimal places for m/z values should suffice.

Reviewer #2 (Remarks to the Author):

I thank the authors for making a series of careful revision to their manuscript in response to the prior round of critiques. I have nothing further to add but a minor remark on semantics (line 409): “The reduced pancreas and islet beta cell mass of CR mass suggested that CR induces a post-mitotic cell state.” Perhaps the post-mitotic state is the default, and it is the lack of restriction/ad lib feeding that keeps some beta cells from staying in a post-mitotic state. The authors are free to ignore this remark if they so choose.

Reviewer #3 (Remarks to the Author):

dos Santos et al. have made a significant efforts to improve their manuscript in response to reviewers. They have successfully addressed my concerns or reasonably justified when it was not possible to do so. Overall, they have strengthened their results by including new experiments using caloric dilution as a caloric reduction without fasting strategy, and doing a diet switch experiment to investigate if CR protected from future dietary challenges. They have as well modified the title and some parts in the introduction/discussion to accurately reflect their findings.

There are just a few minor changes in some figures that should be addressed:

- In figure 4D, the legend for the dot size is missing.
- In the figure legend of figure 6E, definition of the p value corresponding to asterisks is missing.

Other than that, I consider the manuscript suitable for publication.

REVIEWER COMMENTS

Reviewer #1 (Remarks to the Author):

The authors addressed the majority of issues me and other reviewers have raised. The quality of the images has been greatly improved. Some minor clarifications should be warranted as pointed out below.

Thank you.

#1. In the section of “MALDI MS metabolomics”, the term “enriched” can be ambiguous. Does this imply a significant upregulation? and #3. Could the author provide more details on the process of ROC curve analysis and the threshold setting?

Yes, it does. We have rewritten the methods section to add more clarity regarding the threshold used to identify significantly enriched metabolites in AL and CR samples. It now contains the following paragraph:

“Images were analyzed with flexImaging and SCiLS Lab software (Bruker) using the Receiver operating characteristic (ROC) method with automatic peak detection to identify a total of n=290 metabolites from the combined mass spectra. An area under the curve threshold of 0.5 ± 0.1 was used to define metabolites that were enriched in CR (n=52 metabolites) or in AL (n=28) islet regions, respectively (Figure 4C, Supp Tables 8-9).”

#2. Line 487, normal levels of metabolites are not suggested to show metabolic alternations.

This sentence has been revised to: *“First, MALDI MS metabolomics reveals elevated levels of lipids and signaling molecules, cAMP, glucose metabolites, and ADP-ribose in CR beta cells – the later which could explain why CR beta cells have lower DNA damage marker expression⁷⁴ (Figure 5).”*

#4. In Figure 5E, the AL enriched part, could the author specify the compound or type of the metabolite ions m/z 327.23271 and 766.53786? Based on the accurate mass, m/z 327.23271 is more likely a fatty acid rather than phosphatidylethanolamine.

We provide an additional supplementary table with all the possible metabolic identifiers based on HMDB query, including both ions mentioned above. This table was removed by mistake during the revision processed and has now been reincorporated into the manuscript as Table S9. Based on the reviewers point, we've decided to reclassify ion 327 as an “unidentified molecule” and ion 766 as a “Unknown fatty acid”. The figures and text have been updated accordingly.

#5. In Figure 5E, please enhance the descriptions in the legend to clearly indicate the locations of islets and acinar compartments. In addition, some professional terminology need to be improved. The expression of “766.53786 m/z \pm 0.00565 Da” is inappropriate, as m/z is a ratio without units and Da is a unit of atomic or molecular mass. Five ions are identified with same $\Delta m \pm 0.00565$ Da. Please verify this. If this represents the m/z tolerance, it can only retain one significant digit. Usually it is allowed not to be indicated in the figures.

#6. The author should provide information about the mass spectrometry resolution in MALDI MS experiments; typically, retaining four decimal places for m/z values should suffice.

Islet regions are now indicated by white arrows. The description of mass values and molecular mass tolerance are imprinted on the images within the SCiLS software package. Therefore, we are unable to remove it to accommodate the reviewers request regarding the terminology without further editing the MALDI figure panels. The methods section has been updated to include the mass resolution”

“Ion mass resolution was of ± 0.00565 Da”.

Reviewer #2 (Remarks to the Author):

I thank the authors for making a series of careful revision to their manuscript in response to the prior round of critiques. I have nothing further to add but a minor remark on semantics (line 409): “The reduced pancreas and islet beta cell mass of CR mice suggested that CR induces a post-mitotic cell state.” Perhaps the post-mitotic state is the default, and it is the lack of restriction/ad lib feeding that keeps some beta cells from staying in a post-mitotic state. The authors are free to ignore this remark if they so choose.

Thank you for your suggestion and we agree with your comment. This section has been slightly revised to:

“The reduced pancreas and islet beta cell mass of CR mice suggested that CR is associated with a mostly post-mitotic cell state with reduced expression of beta cell aging markers and enhanced cell and organelle homeostasis (Figures 1-7)”

Reviewer #3 (Remarks to the Author):

dos Santos et al. have made a significant efforts to improve their manuscript in response to reviewers. They have successfully addressed my concerns or reasonably justified when it was not possible to do so. Overall, they have strengthened their results by including new experiments using caloric dilution as a caloric reduction without fasting strategy, and doing a diet switch experiment to investigate if CR protected from future dietary challenges. They have as well modified the title and some parts in the introduction/discussion to accurately reflect their findings.

Thank you.

There are just a few minor changes in some figures that should be addressed:

- In figure 4D, the legend for the dot size is missing. *This figure has been re-drawn to highlight the gene names and a more detailed figure legend has been added.*
- In the figure legend of figure 6E, definition of the p value corresponding to asterisks is missing.

This information was originally shown on the graph itself (same for panel G). We have added a p value definition in the legend as well: “For (E) and (G), the p values are shown below the line graph”.

Other than that, I consider the manuscript suitable for publication.

REVIEWERS' COMMENTS

Reviewer #1 (Remarks to the Author):

Thanks for considering the reviewer's suggestions. I believe the current version is suitable for publication.